# Interdisciplinary interventions that improve patient-reported outcomes in perioperative cancer care: A systematic review of randomized control trials

**Bhagvat J. Maheta**[1,2], **Nainwant K. Singh**[1,3], **Karl A. Lorenz**[1,4], **Sarina Fereydooni**[5], **Sydney M. Dy**[6], **Hong-nei Wong**[7], **Jonathan Bergman**[8,9], **John T. Leppert**[1,10], **Karleen F. Giannitrapani**[1,4]*

1 VA Center for Innovation to Implementation, Menlo Park, CA, United States of America, 2 California Northstate University College of Medicine, Elk Grove, CA, United States of America, 3 Department of Health Policy, Stanford University School of Medicine, Stanford, CA, United States of America, 4 Department of Primary Care and Population Health, Stanford University School of Medicine, Stanford, CA, United States of America, 5 Yale University, New Haven, CT, United States of America, 6 Johns Hopkins Bloomberg School of Public Health, Baltimore, MD, United States of America, 7 Lane Medical Library, Stanford University School of Medicine, Stanford, CA, United States of America, 8 VA Los Angeles Healthcare System, Los Angeles, CA, United States of America, 9 Olive View UCLA Medical Center, Los Angeles, CA, United States of America, 10 Department of Urology, Stanford University School of Medicine, Stanford, CA, United States of America

* karleen@stanford.edu

**Data Availability Statement:** All relevant data are within the paper and its Supporting Information files.

## Abstract

### Introduction

Interdisciplinary teams are often leveraged to improve quality of cancer care in the perioperative period. We aimed to identify the team structures and processes in interdisciplinary interventions that improve perioperative patient-reported outcomes for patients with cancer.

### Methods

We searched PubMed, EMBASE, and CINAHL for randomized control trials published at any time and screened 7,195 articles. To be included in our review, studies needed to report patient-reported outcomes, have interventions that occur in the perioperative period, include surgical cancer treatment, and include at least one non physician intervention clinical team member: advanced practice providers, including nurse practitioners and physician assistants, clinical nurse specialists, and registered nurses. We narratively synthesized intervention components, specifically roles assumed by intervention clinical team members and interdisciplinary team processes, to compare interventions that improved patient-reported outcomes, based on minimal clinically important difference and statistical significance.

### Results

We included 34 studies with a total of 4,722 participants, of which 31 reported a clinically meaningful improvement in at least one patient-reported outcome. No included studies had

**Funding:** Dr. Karleen Giannitrapani is supported by a VA Career Development Award (19-075). The full name of the funder is the Veterans Affairs and this is the link to the funder website: https://www.research.va.gov/funding/cdp.cfm. Dr. Nainwant Singh is supported by an Advanced Fellowship in Health Services Research sponsored by the Elizabeth Dole Center of Excellence and VA Center for Innovation to Implementation. The full name of the funder is the Veterans Affairs and this is the link to the funder website: https://www.hsrd.research.va.gov/centers/dole/ The funders had no role in study design, data collection and analysis, decision to publish, or preparation of the manuscript.

**Competing interests:** The authors have declared that no competing interests exist.

an overall high risk of bias. The common clinical team member roles featured patient education regarding diagnosis, treatment, coping, and pain/symptom management as well as postoperative follow up regarding problems after surgery, resource dissemination, and care planning. Other intervention components included six or more months of continuous clinical team member contact with the patient and involvement of the patient's caregiver.

## Conclusions

Future interventions might prioritize supporting clinical team members roles to include patient education, caregiver engagement, and clinical follow-up.

## Introduction

Teams play an important role in fostering better care for patients with serious conditions or when treatment is complex [1]. A team can be either bounded, where members have static, defined roles, or dynamic, where members adjust roles continuously based on the task [2, 3]. Traditionally, nurses and physicians had their own individual roles when treating patients, and their responsibilities did not overlap or change frequently while treating a patient [2]. The dynamic model of teaming is important in healthcare due to the dynamic nature of patient disease trajectories and medical care generally, allowing different members of the healthcare team to adjust their roles as needed to best take care of the patient [4, 5]. Since physicians often face time constraints when caring for patients, interdisciplinary teams also engage with patients and families to address care needs [6, 7]. Interdisciplinary teams can be defined as a team that synthesizes and harmonizes links between each individual discipline into a coordinated and coherent whole [8]. Leveraging non-physician clinical team members (e.g. advanced practice providers, clinical nurse specialists, and registered nurses) to participate on interdisciplinary teams can improve the quality of life for patients, especially in the context of cancer care [9–11].

Both cancer and its treatment often impairs aspects of quality of life (e.g. physical, emotional, existential, and social); because patient needs fluctuate over time, an interdisciplinary team can be leveraged to support patients' changing needs so that the care is customized for each patient [5, 10–11]. Recent interventions in perioperative cancer care have revolved around integrating interdisciplinary teams to improve patient education, follow-up, and rehabilitation, leading to improved patient outcomes [10]. For example, in a randomized control trial where a nurse practitioner joined the oncology team to discuss hospice, living wills, and advanced directives with patients metastatic cancer, there was an improvement in the patients' quality of life, overall physical, mental, and emotional wellbeing [11]. Developing opportunities to improve the quality of life for patients with cancer is particularly important because the symptoms of cancer, as well as the treatment, such as surgery, chemotherapy, or radiation therapy, can severely diminish a patient's quality of life [12]. Although surgery can help treat cancer and alleviate symptoms, surgery itself can lead to adverse side effects such as loss of function, pain, and fatigue [13]. Improving the quality of life of patients with cancer is important, both due to the adverse effects of cancer, and its treatment.

The perioperative period is an exemplary context in which teams may play an important role in fostering better outcomes in cancer care [14, 15]. During the perioperative period, surgeons and surgical teams have many complex tasks which require interdisciplinary collaboration: (1) communicate the purpose and expect results of surgery (which may be curative, life-

extending, or palliative), (2) identify patients who are appropriate for surgery based on relative risk of frailty or life expectancy, (3) optimize patients for surgery to minimize risks, and (4) quickly identify and manage complications postoperatively [15–18].

These tasks are critical to complete during the perioperative period since frailty, which is highly prevalent in the cancer population, can be associated with increased postoperative morbidity and mortality, which is an indicator of poor quality of care [17, 18]. The risks faced by surgical cancer patients over time is underscored by the fact that over 20% of cancer patients over the age of 70 die within 5 years of surgery [12]. Since there is an increased incidence of morbidity and mortality in the perioperative period of cancer care, there is an opportunity for leveraging diverse clinical team members to improve outcomes for patients within this time period [10, 18–20].

Through this systematic review, we sought to understand how clinical team members might improve perioperative care and how their roles might be further modified or extended to improve perioperative care quality. Despite a recent surge in interventions integrating interdisciplinary teams in perioperative cancer care, individual components of interventions, such as specific structures and processes have not been analyzed. The purpose of this paper was to identify the structures (who is on the team and what are their roles) and processes (how the team functions and communicates) in existing interdisciplinary interventions that improved perioperative patient reported outcomes (PROs) among patients with cancer in randomized controlled trials to garner insights on how future interventions might improve PROs in perioperative cancer care. We also aimed to understand the various components of interventions and the impact they had on PROs. This includes the types of interventions, specific team member roles, and caregiver involvement.

## Methods

We used the Preferred Reporting Items for Systematic Reviews and Meta-Analysis Protocols (PRISMA-P) Statement and EQUATOR guidelines to guide the creation of this protocol [21, 22]. We registered this study protocol to the PROSPERO database under the registration number CRD42021270688 [23]. We aimed to identify the structures and processes in interdisciplinary interventions that improve perioperative PROs in randomized controlled trials for patients with cancer. For this paper, we define the non-physician clinical team members as advanced practice providers, clinical nurse specialists, and registered nurses. PROs of interest included the Quality of Life Scales (QLQ-c30), Hospital Anxiety and Depression Scales (HADS), Symptom Distress Scale (SDS), 12-Item Short Form Health Survey (SF-12), Functional Assessment of Cancer Therapy (FACT-C), Functional Assessment of Cancer Therapy (FACT-B), Self-rating Anxiety Scale (SAS), Self-rating Depression Scale (SDS), Profile of Mood States, Gracely Pain Scale, Center for Epidemiologic Studies Depression Scale, Stoma Self-Efficacy Scale, Numerical Rating Scale for pain intensity, American Pain Society- patient questionnaire, and the Mood Adjective Check List (MACL). These metrics help us better understand the impact of interventions on a person's life.

### Study selection

To search for the articles that fit the scope of this study, we consulted a professional research librarian (HW). Key concepts included in the search strategy included randomized control trials, cancer care, and clinical team members as shown in **Appendix S1 (S1 File)**. We searched PubMed, Embase, and CINAHL in June 2021 and again to update the search in March 2023 for studies published at any time and ended up with 11,222 articles (8,020 articles after duplicates were removed). *Inclusion criteria* were: RCTs, adults (18 and over), English only (for

feasibility), PROs, inclusion of clinical team members as part of the healthcare team, perioperative period (defined as 30 days before and up to 90 days after surgery), and cancer care in the in-patient or out-patient/ ambulatory setting [15]. *Exclusion criteria* were: non-RCTs, protocol papers, pediatric population, studies not written in English, no PROs, no clinical team member in the intervention, not during the perioperative period, no patients with cancer, or interventions testing the technical execution of surgery, radiation, pharmaceutical delivery, or other clinical therapies.

We also excluded studies of art/music therapy, exercise therapy, diet changes, or spiritual therapy for patients because these interventions have been extensively studied in the literature as effective programs on their own. These represent specific clinical interventions that have been shown to be effective on their own, regardless of what team members facilitating the intervention, and we wanted this systematic review to focus on the components of interventions (e.g. team structure and processes) that impacted PROs [24–26]. The eligibility criteria are detailed through the population, intervention, comparator, outcomes, timing, and setting (PICOTS) framework as shown by Table 1.

During the title/abstract screening and full-text screening phases, two reviewers evaluated each study and were blinded to each other's decisions (BM and SF). During the title/abstract screening (inter-rater reliability; Cohen's Kappa = 0.94022), we resolved all conflicts by a group consensus or by a "gold standard" reviewer (KG). We screened the full texts that remained after the title/abstract screening in a similar way. During the full-text screening (inter-rater reliability; Cohen's Kappa = 0.84632), the reason for exclusion was also identified. Only the "gold standard" reviewer resolved conflicts during the full-text screening phase (KG). We used the Covidence software to generate a PRISMA diagram to track the studies at each stage of the review [27]. During this process, if any systematic reviews met our inclusion criteria, we added their references lists to the title/abstract screening process. We then rejected these systematic reviews that met inclusion criteria, but kept any additional articles gained through screening of the reference lists.

**Table 1. PICOTS eligibility criteria.**

| | |
|---|---|
| **Population** | **Inclusion** <br> • Adult patients receiving cancer care in the perioperative period in both the control and treatment groups. |
| | **Exclusion** <br> • Studies that include pediatric/ adolescent patients. |
| **Intervention** | **Inclusion:** <br> • We will include studies that have a clinical team member that is trained for the study and provides the intervention for the patient. |
| | ○ This includes advanced practice providers, clinical nurse specialists, or registered nurses. |
| | ■ Advanced practice providers include physician assistants, nurse practitioners, certified nurse midwives, and certified registered nurse anesthetists. |
| | **Exclusion** <br> • All studies exclusively examining the technical execution of surgery, radiation, or pharmaceutical delivery itself will not be included in the study. |
| | • All studies exclusively examining the technical execution of art/music therapy, exercise therapy, diet changes, or spiritual therapy will be excluded. |
| **Comparator** | • Usual care |
| **Outcomes** | • Any patient reported outcomes |
| **Timing** | • Interventions with any follow-up period will be included. |
| **Setting** | • Any care setting (including in-person surgical clinics or out-patient/ ambulatory care) as long as the intervention encompasses the surgical team and perioperative care. |

## Data collection

During the screening process, we built an abstraction form through an iterative process which included palliative care physicians (NS, KL, and SD) as well as practicing urologists (JB and JL). All parts of the intervention were recorded, including what the intervention entailed, the roles of the clinical team members, team structures, team processes, and the extent of caregiver involvement. To expand upon the current knowledge of team based interventions, the definition of teamwork was broadened to include teaming. In the search strategy, "team" or "teamwork" or "teaming" was not used as a limiting term since articles that utilize teaming within the healthcare team in the intervention might not explicitly mention these words. To ensure the processes by which teaming occurred in the intervention was collected, the creation of the abstraction guide was informed by the broadened definition. In the abstraction guide, the definition for teaming used was at least two clinicians interacting with each other and with patients over at least two distinct timepoints [28].

The main outcomes collected were any form of PROs. PROs in the perioperative period can be used to quantitatively assess a patient's quality of life [29]. Healthcare teams can utilize PROs as tools to improve patient outcomes, inform best practices, and refine patient care [30]. Through different patient-reported outcomes (such as the Functional Assessment of Cancer Therapy Scale, QLQ-c30, Profile of Mood States, MACL, HADS), researchers can understand how study interventions may impact patient quality of life [29].

Additional outcomes we collected were evidence of family involvement in decision making, whether the intervention took place during the perioperative period (defined as 30 days before and up to 90 days after surgery) [16]. The entire abstraction guide is shown in **Appendix S2 (S1 File)**. Data abstraction for the included studies was done similar to the screening process in which two reviewers (BM, NS, or SF) abstracted each article independently and resolved all conflicts through a consensus conversation. Risk assessment for bias (**Appendix S3 in S1 File**) was also done in the same way using the Cochrane Risk of Bias tool for randomized control trials [31]. The Cochrane Risk of Bias tool is an accepted risk of bias assessment for RCTs and considered seven domains: sequence generation, allocation concealment. blinding of participants and personnel, blinding of outcome assessors, incomplete outcome data, selective outcome reporting, and other sources of bias [31].

## Data synthesis

We performed a narrative synthesis because of the high degree of heterogeneity stemming from the different forms of PROs used in the studies. Each included study used different methods, and reported different outcomes, which hinders meaningful statistical pooling. We evaluated both clinically significant differences and statistically significant differences. Specifically, we considered the intervention components and mapped them to clinically meaningful improvement in PROs defined by the minimal clinically important difference (MCID) [29, 32]. MCID is defined as the smallest possible change in a reported outcome that has a clinical impact [29, 32]. In studies with PROs that as of yet do not have established MCID, we only considered statistical significance since this was the only reliable metric available to analyze the data from the included studies. The entire intervention description provided was compared to the other intervention descriptions to find similar components across the various interventions. We analyzed the expanded roles of clinical team members and additional components of interventions that were done by the design of the intervention on top of the usual care provided to the control group for the respective article. For articles where the control group received usual care and additional care such as physical therapy sessions, we considered the roles and intervention components that were added on top of the control. Different clinical

team member roles, as well as intervention components, were then categorized based on if they showed a clinically meaningful improvement in PROs compared to the control group in their respective studies. We used this to extrapolate results and recommendations to improve perioperative cancer care from these sparse studies. To ensure the accuracy and reliability of the synthesis, all intervention components extrapolated were discussed and modified through an interactive process with the entire team. The roles and intervention components were defined as follows:

- *Common roles* = Roles taken on by the clinical team members during the intervention [33].

- *Group education* = Formal educational sessions where groups of patients got together to learn from the healthcare team [34].

- *Patient/caregiver education* = One on one education with the patient without a formal large group education session [35, 36].

- *Clinical follow-up* = Standard follow-up with patients where symptoms are assessed, advice was given, and any questions are answered. The primary purpose of the follow-ups was not to solely educate the patients [37].

- *Contact with patient throughout intervention* = Contact throughout the intervention means that the clinical team member met with the patient in some capacity at least two different times throughout the intervention.

- *Team structures* = Who was part of the team, their roles, and their responsibilities.

- *Team processes* = Any form of communication/ collaboration that occurred with the inter-disciplinary healthcare team. (eg. whether they had consistent meetings, ways they communicated, etc. . .) [38].

- *Consistent communication* = Consistent communication includes any communication/ collaboration processes that were set up among the interdisciplinary healthcare team. This included having consistent meetings and meeting with other team members at least two different times throughout the intervention.

- *Referrals* = The clinical team member directed the patient to other professionals such as physicians, other members of the interdisciplinary healthcare team, physiotherapists, psychologists, and nutritionists.

- *Caregiver involvement* = Caregivers included family members, friends, or surrogate decision makers who were somehow included in the intervention [39].

## Results

### Literature selection

In total, 8,020 titles/abstracts were screened based on the inclusion and exclusion criteria. 472 full-text articles were then screened for eligibility as shown by **Fig 1**. We included 34 total studies in our review [40–73], of which 31 reported a clinically meaningful improvement in at least one PRO [40, 42–44, 46–58, 60–73]. The included articles are summarized in **Table 2** and **Appendix S4 (S1 File)**. The studies reported 15 different PROs, as shown in **Appendix S5 (S1 File)**. **Appendix S5 (S1 File)**also provides the ranges for the included PROs that determine MCID based on the most recent published data available. Team structures varied vastly across the interventions, wrangling from a nurse and physician team [40, 45, 50, 54, 56, 57], a nurse and another member of the interdisciplinary healthcare team [43, 52, 60, 66], and teams

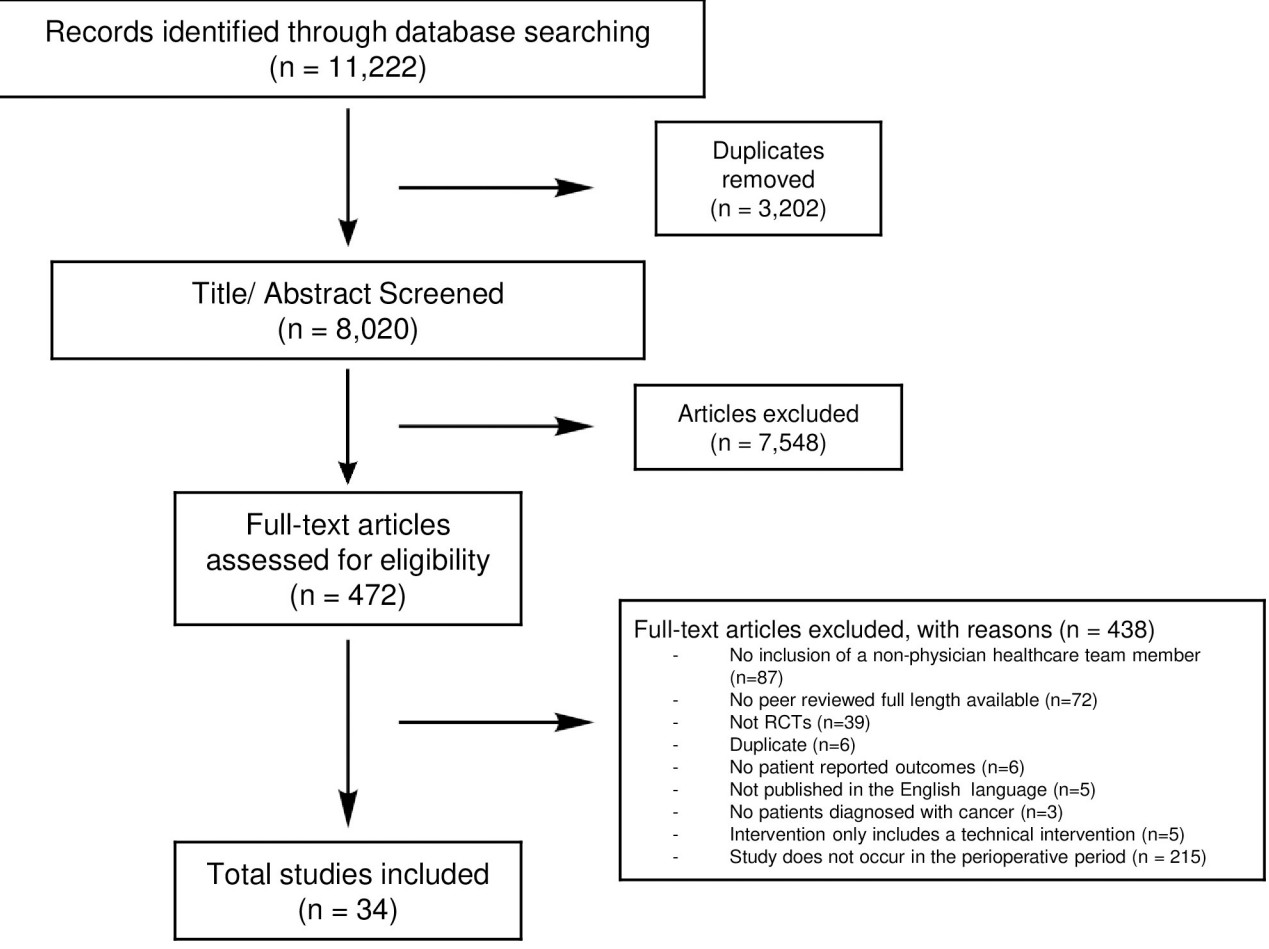

**Fig 1. PRISMA flowchart** [74].

including three or more interdisciplinary healthcare team members [41, 42, 44, 47–49, 55, 61, 63, 64, 71]. Team processes included consistent communication within the interdisciplinary healthcare team at multiple different points in time [40, 44, 45, 51, 56, 62, 64, 67, 71]. Intervention components included clinical team members roles of group education, patient/caregiver education, and clinical follow-up, contact with the patient throughout the intervention, and caregiver involvement. Three studies included advanced practice providers (one article with advanced practice nurses and two articles with nurse practitioners) [43, 44, 55], nine studies included clinical nurse specialists [40–42, 45–49, 60], and 23 studies included registered nurses [50–59, 61–73] (Wallen et al., 2012 included both an advanced practice provider and a registered nurse in their intervention); all three studies with advanced practice providers reported a clinically meaningful improvement. No included studies had an overall high risk of bias as per the Cochrane Risk of Bias tool.

## Common roles

Common roles taken on by the clinical team members throughout the interventions included group education, patient/caregiver education, and clinical follow-up. The roles, interventions, and PROs are summarized in **Table 3**. The education roles revolved around providing patients with more information regarding their diagnosis, potential complications, pain management,

**Table 2. Summary of the 27 included studies.**

| Study | Patient Characteristics | Intervention Context | Clinical team member (advanced practice provider, clinical nurse specialist, or registered nurse) | Common Roles | Contact with Patient Throughout Intervention | Team Processes | Caregiver Involvement | Patient-reported Quality Outcomes |
|---|---|---|---|---|---|---|---|---|
| Inman et al., 2011 | 60 males (average age: 61.25) with prostate cancer undergoing a radical prostatectomy. | The intervention started before the surgery and continued with a telephone call 3–5 days after the surgery. | Clinical Nurse Specialists | Participated in educational classes teaching patients about radical prostatectomies | Telephone and in-person follow-up before and after the surgery | Interdisciplinary team held multiple meetings to create the packet for the educational classes | Caregivers participated in the educational class | Improvement in patient reported healthcare knowledge |
| Koinberg et al., 2004 | 264 females (average age: 59.4) with breast cancer undergoing a mastectomy or partial mastectomy | The intervention started 3 months after surgery (at the end of the perioperative period) and continued afterwards. | Clinical Nurse Specialists | Educated patients in identifying recurrence of cancer and gave advice on self care | Telephone and in-person contact with patients at 1 year intervals | Referrals to physiotherapists and physicians | No | No improvement in Hospital Anxiety and Depression scale (HADS) |
| Malmstrom et al., 2016 | 82 patients (average age: 66.45) with esophageal cancer undergoing an esophagectomy or esophagogastrectomy. | The intervention started immediately after surgery during discharge and continued afterwards. | Clinical Nurse Specialists | Followed up with patients regarding life after surgery, self-care, plans for the future, and additional resources available if needed | Telephone and in-person contact 16 times over the course of 6 months | Referrals to nutritionists, physiotherapists, and surgeons | No | Improvement in Quality of Life Scales (QLQ-c-30) |
| McCorkle et al., 2009 | 123 females (average age: 60.25) with ovarian cancer undergoing gynecological surgery. | The intervention started immediately after surgery during discharge and continued afterwards. | Advanced Practice Nurses | Educated patients on symptom management, diagnosis/ treatment, and provided support via counseling | Telephone and in-person contact for 6 months | Referrals based on screening to psychiatric consultation–liaison nurses | Advanced practice nurses talked to families about management strategies and involved them in discussion with the patient | Improvement in Symptom Distress Scale (SDS) and 12-Item Short Form Health Survey (SF-12) |
| Koet et al., 2021 | 75 patients (average age: 71.5) with colorectal cancer undergoing laparoscopy or laparotomy. | The intervention occurred 1 week prior to surgery. | Nurse Practitioners | Gave PowerPoint presentation during an educational class teaching patients about their diagnosis, therapy, hospitalization, potential side effects, functional implications and provided psychoeducation to help patients cope with cancer | No | Team met together to develop the training session | Caregivers participated in the educational class | Improvement in Quality of Life Scales (QLQ-c30) |

*(Continued)*

**Table 2.** (Continued)

| Study | Patient Characteristics | Intervention Context | Clinical team member (advanced practice provider, clinical nurse specialist, or registered nurse) | Common Roles | Contact with Patient Throughout Intervention | Team Processes | Caregiver Involvement | Patient-reported Quality Outcomes |
|---|---|---|---|---|---|---|---|---|
| Verschuur et al., 2009 | 109 patients (average age: 61) with esophageal or gastric cardia cancer undergoing curative surgery. | The intervention started after surgery and continued afterwards. | Clinical Nurse Specialists | Followed up with patients regarding their health care problems and symptoms, their body weight, and any dysphagia issues due to their esophageal cancer | Telephone and in-person contact for 12 months | Held 4 weekly interdisciplinary team meetings to discuss patients | No | No improvement in Quality of Life Scales (QLQ-c30) |
| Mertz et al., 2017 | 50 females (average age: 52.95) with breast cancer undergoing mastectomy or breast-conserving surgery. | The intervention started immediately after surgery during discharge and continued afterwards. | Clinical Nurse Specialists | Educated patients on pain management, preventing nausea/ vomiting, and temporary breast prosthesis. Also provided psychological support through counseling. | Two in-person and 2–7 telephone conversations with the patient. | Referrals to psychologists. | No | Improvement in Quality of Life Scales (QLQ-c30) and Hospital Anxiety and Depression Scales (HADS) |
| Sussman et al., 2017 | 193 patients (average age: 60.46) with breast or colorectal cancer undergoing surgical treatment. | The intervention started immediately after surgery during discharge and continued afterwards. | Clinical Nurse Specialists | Educated patients on coping with and managing the effects of cancer and provided emotional support | Telephone and in-person contact at least two times (one week apart) | Referrals to other cancer care services | No | Improvement in Quality of Life Scales (QLQ-c30) |
| Li et al., 2016 | 226 females (average age: 46.11) with early stage cervical cancer undergoing surgery. | The intervention started immediately after surgery during discharge and continued afterwards. | Clinical Nurse Specialists | Led educational class that taught physiological rehabilitation, emotional release management and provided a space for informal social support systems | Telephone contact every two weeks and home visit every 2–3 months for 6 months | No | Social support systems in educational class | Improvement in Quality of Life Scales (QLQ-c30) |
| Harrison et al., 2011 | 74 patients (average age: 64.57) with colon, rectum, or rectosigmoid cancer undergoing a colectomy, anterior resection, or abdominoperineal resection. | The intervention started immediately after surgery during discharge and continued afterwards. | Clinical Nurse Specialists | Followed up with patients regarding their physical, psychosocial, information, supportive care, and rehabilitation needs | Telephone and in-person contact for 6 months | Referrals to rest of clinical team | Caregivers and family members could act as interpreters if needed during telephone calls | Improvement in Functional Assessment of Cancer Therapy (FACT-C) |

*(Continued)*

**Table 2.** (Continued)

| Study | Patient Characteristics | Intervention Context | Clinical team member (advanced practice provider, clinical nurse specialist, or registered nurse) | Common Roles | Contact with Patient Throughout Intervention | Team Processes | Caregiver Involvement | Patient-reported Quality Outcomes |
|---|---|---|---|---|---|---|---|---|
| Quist et al., 2018 | 211 patients (average age: 65.52) with non-small cell lung cancer undergoing Video-assisted thoracoscopic surgery (VATS) or open surgery. | The intervention started after the surgery and continued afterwards. | Clinical Nurse Specialist | Led educational sessions that included group exercise, individual counseling, and health-promoting behaviors | In-person contact for 12 weeks | No | No | Improvement in Quality of Life Scales (QLQ-c30) |
| Wallen et al., 2012 | 152 patients (average age: 52.41) with advanced mesothelioma, mucinous adenocarcinoma, melanoma, adenocarcinoma, ocular melanoma, or colon cancer undergoing surgery. | The intervention started immediately after the surgery. | Nurse Practitioners and Registered Nurses | Followed up regarding physical symptoms, and emotional/ spiritual distress | No | Referrals to physicians, spiritual ministry, social work, recreation therapy, counseling, nutrition, acupuncture, massage, and rehabilitation medicine | No | Improvement in Gracely Pain Scale, Symptom Distress Scale, and Center for Epidemiologic Studies Depression Scale |
| Zhou et al., 2020 | 111 females (average age: 49.91) with breast cancer undergoing a modified radical mastectomy, total mastectomy, or breast-conserving surgery. | The intervention started when the patient was admitted to the hospital and continued after the surgery. | Registered Nurses | Followed up with patients and assisted them in physical rehab, psych rehab, social rehab and provided rehabilitation information | WeChat and in-person contact for 6 months | No, but the registered nurses, physicians, and research team worked together to deliver the intervention itself | Family/ spouse accompanied the patient as much as possible | Improvement in Health-Related Quality of Life via FACT-B |
| Zhu et al., 2021 | 81 patients (average age: 56.15) with gastric cancer (in gastric fundus/ cardia, gastric body, or gastric antrum) undergoing a proximal gastrectomy, distal gastrectomy, or total gastrectomy. | The intervention started immediately after the surgery. | Registered Nurses | Followed up with patients to manage patients pre-op, intr-op, and post-op. | In-person contact for 3 days, which was the duration of the intervention | The nurses each had their own roles and coordination was led by the head nurse | No | Improvement in Self-rating Anxiety Scale (SAS) and Self-rating Depression Scale (SDS) |
| Zheng and Jiang, 2019 | 86 patients (average age: 51.85) with lung cancer undergoing surgery. | The intervention started immediately after the surgery. | Registered Nurses | Provided health education to the patient and informed them about healthy diets, exercise, and quitting smoking and alcohol use | In-person contact for 2 days, which was the duration of the intervention | No | No | Improvement in 30-Item Short Form Health Survey (SF-30) |

(*Continued*)

**Table 2.** (Continued)

| Study | Patient Characteristics | Intervention Context | Clinical team member (advanced practice provider, clinical nurse specialist, or registered nurse) | Common Roles | Contact with Patient Throughout Intervention | Team Processes | Caregiver Involvement | Patient-reported Quality Outcomes |
|---|---|---|---|---|---|---|---|---|
| Young et al., 2013 | 756 patients (average age: 67.82) with colon, rectal, or stoma cancer undergoing surgery. | The intervention started 3 days after surgery and continued afterwards. | Registered Nurses | Followed up with patients to screen for psychosocial, physical, information, supportive care, and rehabilitation needs | Telephone and in-person contact for 6 months | Referrals to local healthcare team | No | No improvement in Functional Assessment of Cancer Therapy (FACT-C), but consistently positive quantitative questionnaire responses |
| Watson et al., 1988 | 40 females (average age: 60) with breast cancer undergoing surgery. | The intervention started after the patient was admitted to the hospital prior to surgery and continued afterwards. | Registered Nurses | Followed up with patients to provide emotional support, help with adjustment, give information about their physical state, and give practical advice on breast prostheses. | In person contact for 2–3 weeks | No | No | Improvement in Profile of Mode States |
| Temur and Kapucu, 2019 | 61 females (average age: 46.58) with breast cancer undergoing modified radical mastectomy or breast care surgery. | The intervention started within the first 24 hours of the surgery and continued afterwards. | Registered Nurses | Followed up with patients and worked with physicians to adjust the patient's exercise regimen | Telephone contact for 6 months | Registered nurses and physicians met to adjust the patient's exercise regimen | No | Improvement in Quality of Life Scales (QLQ-c30) |
| Sui et al., 2020 | 200 patients (average age: 61.86) with non-small cell lung cancer undergoing surgical resection. | The intervention started after the surgery and continued afterwards. | Registered Nurses | Followed up with patients to provide education, rehabilitation, exercise guidance, daily activity supervision, and psychological support | Telephone, WeChat, and in-person contact for 12 months | Referrals to physicians and oncologists | No | Improvement in Hospital Anxiety and Depression Scales (HADS) |
| Xu et al., 2017 | 48 patients (average age: 61.04) with colorectal cancer undergoing an intestinal stoma. | The intervention started after the surgery and continued afterwards. | Registered Nurses | Educated patients through a self efficacy intervention | Telephone and in-person contact for 3 months | No | Family members and friends participated in the rehabilitation course | Improvement in Stoma Self-Efficacy Scale |

(*Continued*)

**Table 2.** (Continued)

| Study | Patient Characteristics | Intervention Context | Clinical team member (advanced practice provider, clinical nurse specialist, or registered nurse) | Common Roles | Contact with Patient Throughout Intervention | Team Processes | Caregiver Involvement | Patient-reported Quality Outcomes |
|---|---|---|---|---|---|---|---|---|
| Ross et al., 2005 | 249 patients (average age: 68.45) with colorectal cancer undergoing radical operation or non-radical operation. | The intervention started after the surgery and continued afterwards. | Registered Nurses | Followed up with patients and provided emotional and informational support | Telephone and in-person contact for 12 weeks | No | Family caregivers received comprehensive clinical assessments, monitoring, and teaching | No improvement in Quality of Life Scales (QLQ-c30) or Hospital Anxiety and Depression Scales (HADS) |
| Boesen et al., 2005 | 241 patients with cutaneous malignant melanoma undergoing surgery. | The intervention started 3 weeks after surgery and continued afterwards. | Registered Nurses | Led educational sessions to provide information about cancer-preventative behavior | In-person contact for 6 weeks | No | No | Improvement in Total Mood Disturbance (TMD) of the Profile of Mood States (POMS) |
| Francke et al., 1997 | 152 patients (average age: 63.81) with colon or breast cancer undergoing resection for colon cancer, mastectomy, or lumpectomy. | The intervention started after the surgery and continued afterwards. | Registered Nurses | Led educational sessions focused on pain, psychosocial interventions, physical/relaxation interventions, and pharmacological pain management | In-person contact for 4 months | Coordination occurred within the team and was led by the coordinating head nurse | No | Improvement in Numerical Rating Scale for pain intensity |
| Allard 2007 | 117 females (average age: 53.6) with breast cancer undergoing surgery. | The intervention started after the surgery and continued afterwards. | Registered Nurses | Followed up with patients to assess symptoms, manage symptoms, provide symptom relief, and suggest self-care strategies | Telephone contact for 1 week, which was the duration of the intervention | No | No | Improvement in Profile of Mood States (POMS) |
| Bjerring et al., 2020 | 79 patients (average age: 71) with esophageal cancer undergoing a surgery to insert a self-expandable metallic stent. | The intervention started after the surgery and continued afterwards. | Registered Nurses | Followed up with patients and covered a standard list of topics as well as any questions/concerns the patients had | Telephone and in-person contact for 11 weeks | Met with interdisciplinary tumor team | Family members assisted in the completion of the respective questionnaires | Improvement in Quality of Life Scales (QLQ-c30) |
| Bahrami et al., 2012 | 70 patients (average age: 44.31) with breast cancer or some form of abdominal cancer undergoing a gastrectomy, mastectomy, laparotomy, lumpectomy, splenectomy, or thoracotomy. | The intervention started before surgery and continued after surgery. | Registered Nurses | Led educational session to teach relaxation techniques and help with pain management | No contact outside of single education session | No | No | Improvement in American Pain Society- patient questionnaire |

(*Continued*)

**Table 2.** (Continued)

| Study | Patient Characteristics | Intervention Context | Clinical team member (advanced practice provider, clinical nurse specialist, or registered nurse) | Common Roles | Contact with Patient Throughout Intervention | Team Processes | Caregiver Involvement | Patient-reported Quality Outcomes |
|---|---|---|---|---|---|---|---|---|
| Ohlsson-Nevo et al., 2017 | 85 patients (average age: 66.01) with colon, rectal, or anal cancer undergoing open surgery or laparoscopic surgery. | The intervention started after the surgery and continued afterwards. | Registered Nurses | Led educational session consisting of discussion groups on colorectal cancer, relaxation, exercise, diet, and dealing with crisis | In-person contact for 7 weeks | No | Caregivers attended a different discussion group than the patients | Improvement in Mood Adjective Check List (MACL) |
| Ding et al., 2022 | 100 patients (average age: not reported) with laryngeal carcinoma undergoing total or partial laryngectomy. | The intervention started after the surgery and continued afterwards. | Registered Nurses | Followed up with patients using the connect, introduce, communicate, ask and respond, and exit sequence they were trained on. | Telephone contact at one, two, four, six, and eight weeks after discharge; thereafter, a follow-up call was made every eight weeks (total of 7 contacts). | All intervention nurses received three training sessions on motivational communication (each session was 1 hour). | No | Improvement in Functional Assessment of Cancer Therapy (FACT)–Head and Neck |
| Hu et al., 2022 | 110 patients (average age: 60.34) with primary hepatic carcinoma undergoing transcatheter arterial chemoembolization. | The intervention started before the surgery and continued intra-operatively. | Registered Nurses | Followed up with patients and provided advice regarding their diet and daily life activities (e.g. improving sleep quality, adhering to medications). | In-person contact throughout the intervention. | No | No | Improvement in 12-Item Short Form Health Survey (SF-12) |
| Ji et al., 2022 | 90 patients (average age: 45.72) with a thyroid tumor undergoing thyroid surgery. | The intervention started before surgery and continued after surgery. | Registered Nurses | Educated patients regarding disease-knowledge to improve patients' awareness of their diseases and enhance the confidence in treatment. Nurses also provided rehabilitation training. | In-person contact throughout the intervention. | No | No | Improvement in Hospital Anxiety and Depression Scales (HADS) |
| Zhao et al., 2021 | 78 patients (average age: not reported) with lung cancer undergoing surgery. | The intervention started before surgery and continued after surgery. | Registered Nurses | Followed up with patients and assessed mental health status, strengthened family and social support, encouraged patients, provided timely feedback, standardized pain management, and provided coping skills. | In-person contact throughout the intervention. | No | Families and friends were encouraged to give more comfort and encouragement to young patients, and the patients were encouraged to communicate with their families and friends. | Improvement in self-rating anxiety scale (SAS) and self-rating depression scale (SDS). |

(*Continued*)

**Table 2.** (Continued)

| Study | Patient Characteristics | Intervention Context | Clinical team member (advanced practice provider, clinical nurse specialist, or registered nurse) | Common Roles | Contact with Patient Throughout Intervention | Team Processes | Caregiver Involvement | Patient-reported Quality Outcomes |
|---|---|---|---|---|---|---|---|---|
| Turkdogan et al., 2022 | 121 patients (average age: not reported) with head or neck cancer undergoing head or neck surgery. | The intervention started before surgery and continued after surgery. | Registered Nurses | Helped create educational videos for the patients to view and assisted them watching the videos and answering any questions. | In-person contact throughout the intervention. | Otolaryngologists worked with a nurse, nutritionist, speech language pathologist, respiratory therapist, physiotherapist, oncology-focused psychologist, and radiation oncologist to design the educational video. | No | Improvement in the European Organisation For Research and Treatment of Cancer Quality of Life Questionnaire (EORTC-QLQ) |
| Yu et al., 2022 | 168 patients (average age: not reported) with esophageal cancer undergoing surgery. | The intervention started after the surgery and continued afterwards. | Registered Nurses | Followed up with patients with telephone calls. Nurses focused on the patients' nutritional status, symptoms after esophagectomy, and any psychological issues, as well as nutritional status and diet counseling. There was also an Internet-based supportive care group on WeChat where nurses would regularly answer patient questions. | Telephone contact once a week for the first two months after discharge, twice a month for months 3–4, and once a month for months 5–6. | No | No | Improvement in the European Organisation For Research and Treatment of Cancer Quality of Life Questionnaire (EORTC-QLQ) |
| Yuan et al., 2022 | 60 patients (average age: 65.22) with hepatocellular carcinoma undergoing hepatic artery interventional chemoembolization. | The intervention started before surgery and continued after surgery. | Registered Nurses | Followed up with patients and provided health guidance regarding diet, symptoms, healthcare instruction guidance, and mental health. | In-person contact throughout the intervention. | No | Yes; family members supported patients and were instructed to urge patients to correct bad habits. | Improvement in the Quality of Life Rating Scale (SF-36), the Numeric Rating Scales (NRS), the Self-Rating Anxiety Scale (SAS), and the Self-Rating Depression Scale (SDS). |

**Table 3. Roles and intervention components associated with improved patient-reported quality outcomes.**

| Roles and Intervention Components | | Improved Patient-reported Quality Outcomes |
|---|---|---|
| **Common Roles** | **Group education:** Formal educational sessions where groups of patients got together to learn from the healthcare team. This included didactic lectures, discussion sections, or a combination of the two [40, 44, 48, 60–62, 65, 66]. | 1. Quality of Life Scales (QLQ-c-30)<br>2. Profile of Mood States<br>3. American Pain Society- patient questionnaire<br>4. Mood Adjective Check List (MACL)<br>5. Numerical Rating Scale for pain intensity |
| | **Patient/caregiver education:** One on one education with the patient without a formal large group education session. This included a formal educational curriculum or a list of topics/ experiences that the patient is taught about [41, 43, 46, 47, 52, 58]. | 1. Quality of Life Scales (QLQ-c30)<br>2. Hospital Anxiety and Depression Scales (HADS)<br>3. Symptom Distress Scale (SDS)<br>4. 12-Item Short Form Health Survey (SF-12)<br>5. Stoma Self-Efficacy Scale |
| | **Clinical follow-up:** Standard follow-up with patients where symptoms are assessed, advice was given, and any questions are answered. The primary purpose of the follow-ups was not to solely educate the patients [42, 45, 49–51, 53–57, 59, 63, 64]. | 1. Quality of Life Scales (QLQ-c30)<br>2. Functional Assessment of Cancer Therapy (FACT-C)<br>3. 12-Item Short Form Health Survey (SF-12)<br>4. Self-rating Anxiety Scale (SAS)<br>5. Profile of Mood States<br>6. Hospital Anxiety and Depression Scales (HADS)<br>7. Symptom Distress Scale (SDS)<br>8. Gracely Pain Scale<br>9. Center for Epidemiologic Studies Depression Scale<br>10. Functional Assessment of Cancer Therapy (FACT-B)<br>11. Numerical Rating Scale for pain intensity |
| **Contact with patient throughout intervention**: The clinical team member met with the patient in some capacity at least two different times throughout the intervention [40–43, 46–50, 53, 56–59, 63, 64]. | | 1. Quality of Life Scales (QLQ-c30)<br>2. Hospital Anxiety and Depression Scales (HADS)<br>3. Symptom Distress Scale (SDS)<br>4. 12-Item Short Form Health Survey (SF-12)<br>5. Functional Assessment of Cancer Therapy (FACT-C)<br>6. Functional Assessment of Cancer Therapy (FACT-B)<br>7. Self-rating Anxiety Scale (SAS)<br>8. Symptom Distress Scale (SDS)<br>9. Profile of Mood States<br>10. Stoma Self-Efficacy Scale<br>11. Mood Adjective Check List (MACL)<br>12. Numerical Rating Scale for pain intensity |
| **Team processes** | **Consistent Communication**: Consistent communication includes any communication/ collaboration processes that were set up among the interdisciplinary healthcare team. This included having consistent meetings and meeting with other team members at least two different times throughout the intervention. [40, 44, 45, 51, 56, 62, 64]. | 1. Quality of Life Scales (QLQ-c30)<br>2. Self-rating Anxiety Scale (SAS)<br>3. Self-rating Depression Scale (SDS)<br>4. Numerical Rating Scale for pain intensity<br>5. Functional Assessment of Cancer Therapy (FACT-C) |
| | **Referrals:** The clinical team member directed the patient to other professionals such as physicians, other members of the interdisciplinary healthcare team, physiotherapists, psychologists, and nutritionists. [41–43, 46–49, 53, 55, 57]. | 1. Quality of Life Scales (QLQ-c30)<br>2. Hospital Anxiety and Depression Scales (HADS)<br>3. Symptom Distress Scale (SDS)<br>4. 12-Item Short Form Health Survey (SF-12)<br>5. Functional Assessment of Cancer Therapy (FACT-C) |
| **Caregiver involvement:** Caregivers included family members, friends, or surrogate decision makers who were somehow included in the intervention [40, 43, 44, 48–50, 58, 59, 64, 66]. | | 1. Quality of Life Scales (QLQ-c30)<br>2. Symptom Distress Scale (SDS)<br>3. 12-Item Short Form Health Survey (SF-12)<br>4. Functional Assessment of Cancer Therapy (FACT-C)<br>5. Functional Assessment of Cancer Therapy (FACT-B)<br>6. Stoma Self-Efficacy Scale<br>7. Hospital Anxiety and Depression Scales (HADS)<br>8. Mood Adjective Check List (MACL) |

relaxation techniques, exercise/diet, psychological support, and preventing cancer in the future. A total of eight studies included educational sessions [40, 44, 48, 60–62, 65, 66] and eight studies included individually educating patients [41, 43, 46, 47, 52, 58, 69, 71], of which only one study did not report a clinically meaningful improvement in at least one PRO. The remaining eighteen studies had clinical follow-up as part of their intervention and eleven of these had a clinically meaningful improvement in at least one PRO [42, 45, 49–51, 53–57, 59, 63, 64, 67, 68, 70, 72, 73]. Interventions with follow-ups may be associated with positive clinical effects on PROs if they included support for a patient's physical, emotional and mental health, psychosocial and spiritual support, and symptom relief.

**Common roles of advanced practice providers.** All three articles that included advanced practice providers in the intervention, regardless of the specific role they performed, were associated with an improvement in PROs compared to the control group in the respective articles [43, 44, 55]. Koet et al., 2021 designed an intervention in which nurse practitioners gave PowerPoint presentations during an educational class to teach patients about their diagnosis, therapy, hospitalization, potential side effects, functional implications and to provide psychoeducation to help patients cope with cancer. The other article that utilized nurse practitioners (Wallen et al., 2012) ensured follow-up with the patients regarding physical symptoms, and emotional/ spiritual distress as part of their intervention. In McCorkle et al., 2009, advanced practice nurses educated patients on symptom management, diagnosis/ treatment, and provided support via counseling.

**Common roles of nursing team members.** The roles taken by clinical nurse specialists were distributed evenly with three articles utilizing interventions where the clinical nurse specialists led educational sessions [40, 48, 60], three articles providing patient/caregiver education [41, 46, 47], and three articles with clinical follow-up [42, 45, 49]. Four articles included registered nurse roles of leading educational sessions [61, 62, 65, 66], while four articles included registered nurse roles of patient/caregiver education with the patient [52, 58, 69, 71], and fourteen articles included registered nurse roles of following up with the patient [50, 51, 53, 54, 56, 57, 59, 63, 64, 67, 68, 70, 72, 73]. Across the studies, there was heterogeneity regarding when the intervention began in the perioperative period relative to the date of surgery, as shown in **Table 2**.

## Contact with patient throughout intervention

In studies where clinical team members maintained contact with patients for over six months throughout the intervention, there was a positive impact on PROs. Twenty-nine of the included studies involved nurse contact with the patient for more than one week [40–43, 45–50, 53, 54, 56–64, 66–73]. Seventeen of these studies used telephone calls to maintain contact with the patient [40–43, 46–49, 53, 56–59, 63, 64, 67, 72] and three of the studies based in China also used WeChat [50, 57, 72]. There was no significant difference between the different platforms for clinical follow-up, whether it was in-person, telephone, or any other modality, suggesting that the general act of contacting the patient throughout the intervention may contribute to improving patient quality of life [40–43, 46–50, 53, 56–59, 63, 64, 67, 72]. The frequency of contact varied across the studies with some interventions having contact every few days, weekly, monthly, every few months, and even yearly. Timing of the clinical team member contact with the patient also varied across the studies in the perioperative period. One study conducted the intervention prior to the surgery [44], twenty-six studies conducted the intervention only after the operation was completed [41–43, 45–49, 51–53, 55–64, 66, 68–70, 73], and seven studies ran their intervention before and after the surgery [40, 50, 54, 65, 67, 71, 72].

## Team structures

Team structures in the intervention arm varied across the 34 different studies. Six studies had a nurse and physician team structure [40, 45, 50, 54, 56, 57] and five of them had improvements in PROs. Four studies had a nurse and another member of the interdisciplinary healthcare team [43, 52, 60, 66] and eleven studies had three or more interdisciplinary healthcare team members [41, 42, 44, 47–49, 55, 61, 63, 64, 71]. Thirteen of the studies only had the nurse as part of the intervention [46, 51, 53, 58, 59, 62, 65, 67–70, 72, 73]. There are not enough included studies, and not enough information regarding team structures in the included studies to draw conclusions about the most effective team structures in improving PROs in perioperative cancer care.

## Team processes

Consistent communication was another component of the interventions that was associated with an improvement in PROs. Nine of the studies included communication within the interdisciplinary healthcare team at multiple different points in time [40, 44, 45, 51, 56, 62, 64, 67, 71]. This included working together to create a training session prior to implementing the intervention together, coordinating with each other throughout the intervention, or meeting with a interdisciplinary team on a frequent basis. Eight of these studies were correlated with improved PROs [40, 44, 51, 56, 62, 64, 67, 71].

Nine of the studies included referrals as a form of communication, and this involved referrals to physicians, the interdisciplinary healthcare team, physiotherapists, psychologists, and nutritionists [41–43, 46–49, 53, 55, 57]. These referrals took place throughout the intervention and were often due to the nurses identifying a factor that could improve the patient's quality of life through referrals. Eight of the interventions that included referrals were associated with an improvement in at least one PRO.

## Caregiver involvement

Involvement of the patient's caregivers may be associated with an improvement in PROs compared to the control group of the respective studies. Caregiver involvement is pertinent because it serves as a way to define quality of care since in the case of serious illness or palliative populations, the patient themselves may have decreased decision making capacity and engagement of the family or caregiver ensures the patient's wishes are respected. Twelve studies included the patient's caregivers, families, social support systems, and/or friends in the intervention and eleven of these studies had at least one clinically significant improvement in PROs [40, 43, 44, 48–50, 58, 59, 64, 66, 70, 73]. Three of the studies allowed caregivers to participate in the educational sessions [40, 44, 48] while some interventions facilitated communication with the caregivers about management strategies and other interventions encouraged caregivers to accompany the patient as much as possible or serve as interpreters.

## Discussion

Multiple randomized controlled trials have evaluated the effect of interdisciplinary team interventions on PROs in the perioperative period, however, the studies have notable heterogeneity. Since no included articles had an overall high risk of bias, all studies were of high quality and thus minimize potential bias in the conclusions drawn from this systematic review. Among the RCTs included, there were common intervention components that might drive a positive clinical effect in PROs. While the specific roles of clinical team members varied across interventions, common roles included leading group education, patient/caregiver education, and

clinical follow-up. Topics for group education included information regarding the diagnosis, relaxation techniques, exercise/ diet, psychological support, and preventing cancer in the future, while follow-up topics included support for a patient's physical, emotional and mental health, psychosocial and spiritual support, and symptom relief. Continuous contact with the patients throughout the intervention (e.g. multiple telephone and in-person follow-ups before and after the surgery) [40] may have a positive impact on PROs in the perioperative period; a majority of included articles included over six months of patient contact. Other components of interventions that may show improvements in PROs were increasing consistent communication and collaboration among the interdisciplinary healthcare team and involving caregivers to support the patient.

In the included studies, clinical team members expanded their roles by facilitating group education, patient/caregiver education, and patient clinical follow-up. Studies have shown that incorporating trained nurses into care processes, with surgeons, can improve cancer care experiences and patient satisfaction determined by PROs [75, 76]. The actual content of the education or follow-up done by the nurses varied vastly and the expansion of the clinical team member role itself serves as a component of interventions that may improve PROs. In the group setting, outcomes in the QLQ-c-30, Profile of Mood States, and MACL were improved, which may be attributed to patients being in the physical presence of each other and providing reassurance and camaraderie, accumulating knowledge together, and asking questions [40, 44, 48, 60–62, 65, 66, 77]. For example, Koet et al., 2021 demonstrated a statistically significant difference in the QLQ-C30 at the one month follow up between the intervention arm (mean = 72.1) and the control arm (mean = 63.9) [44]. The model of patient/caregiver education and clinical follow-up has been well-established in cancer care and personalizes the patient education experience based on their own recovery and allows patients privacy for uncomfortable topics [78–80]. Due to limited resources and time for surgeons, clinical team members can take on an expanded role in cancer care to conduct patient/caregiver education or clinical follow-up with the patients [81–83].

This narrative systematic review adds to the current knowledge of how team based interventions improve PROs in cancer care, which supports potentially expanding the healthcare team by involving additional clinical team members to share the care alongside physicians. Articles that include interventions that leverage clinical team members may not necessarily use the terms "team" or "teamwork" or "teaming" to define the interactions among the clinical team members and the rest of the interdisciplinary healthcare team. In our abstraction form, we captured both formal team based interventions as well as dynamic teaming processes. Broadening the definition of teamwork to also include teaming was necessary so that we could include more articles that leveraged interdisciplinary contributions, which allowed us to demonstrate sharing care with multiple clinical team members is a common intervention feature among articles with improved patient-reported outcomes in perioperative cancer care.

In light of the high rates of physician burnout and limited time and resources for physicians, advanced practice providers can help share the care by serving as full time equivalents (FTEs) for physicians [6–8, 81]. Inclusion of advanced practice providers has been shown to reduce overall cost and provide support for physicians [82]. Given the well-known workforce constraints for palliative care in cancer, advanced practice providers are an ideal alternative to maintain quality of care while considering overall healthcare expenditure [83]. Oncology advanced practice providers can lead follow-up visits and educational sessions for patients in addition to administrative and clinical responsibilities, making this a viable opportunity specifically in the context of perioperative cancer care [84].

Clinical team members contact with patients, and communication with the rest of the interdisciplinary healthcare team at multiple times, are both key components of interventions that

may be associated with improved PROs for patients with cancer in the perioperative period. Longitudinal patient contact is seen in cancer care contexts, where long-term patient contact with physicians or other healthcare professionals is associated with improved patient care [85–87]. In almost all the included perioperative studies, longitudinal patient contact was seen, but was especially exemplified by Mertz et al., 2017 where there were no improvements in patient reported outcomes at six months, but there were clinically and statistically significant improvements in the intervention group at twelve months [46]. In the literature, several studies have shown that using consistent communication within the healthcare team can be an effective part of interventions for treating patients with different types of cancer in different stages of treatment [5, 88, 89]. Longitudinal follow-up and effective communication among the healthcare team, with referrals to healthcare team members, may be used to improve the quality of life for patients with cancer in the perioperative period [90].

In cancer care, many studies have been provider focused rather than involving patient/ caregiver input, so, including caregivers in interventions, and utilizing their input in developing interventions serves as an opportunity for improvement [91]. Involvement of the patient's caregivers in ten of the included articles came in different forms, differing on who was included and how they contributed to the intervention [40, 43, 44, 48–50, 58, 59, 64, 66, 70, 73]. Studies focusing on cancer care have shown the effectiveness of involving caregivers to allow for shared decision making to improve mental health and resilience, which is consistent with improvements in the QLQ-c30, Stoma Self-Efficacy Scale, HADS, and MACL PROs [92, 93]. Including caregivers in cancer interventions has also translated to improved patient satisfaction, suggesting it may be a worthwhile component to include in the treatment of patients with cancer in the perioperative period [94].

Our study can be considered in light of the following limitations. The incorporation of advanced practice providers in the interdisciplinary healthcare team is recent and only three articles that had an intervention with advanced practice providers were included from a limited body of literature. We anticipate an increasing number of publications with interventions that include advanced practice providers. The patient reported outcomes included were heterogeneous and thus a meta-analysis or further synthesis is not possible. Further, PROs hold the bias that patients may have their own individual reference points for answering the questions [95]. Our abstraction form did not capture the type of surgeon, which may be relevant in understanding PROs in different perioperative contexts. Furthermore, the search strategy itself, and the screening process, may have missed certain articles. The included studies were heterogeneous in nature, which did not allow for statistical analysis, thus limiting the conclusions formed in this study. Finally, the included studies did not provide sufficient information in the manuscript regarding the purpose of surgeon interaction with the patient during the perioperative period (communicate the purpose and expect results of surgery, identify patients who are appropriate for surgery, optimize patients for surgery to minimize risks, or identify and manage complications postoperatively), which may have added an additional layer of understanding. Future studies could consider the type of surgeon and the purpose of surgeon interaction with the patient during the perioperative period to better understand if differences in these factors may impact PROs.

## Conclusion

By broadening the definition of teamwork to include teaming, we were able to build upon the current knowledge of team-based interventions, particularly for improving PROs in perioperative cancer care. Our review demonstrated common structures and processes across interventions that impacted PROs in the perioperative period. These included expanding the roles of any clinical team members (advanced practice providers, nurse practitioners, or registered nurses) to include either group education, patient/caregiver education or clinical follow-up

since they all demonstrated improved PROs. Other intervention components included longitudinal follow-up for longer than six months, increasing consistent communication among the interdisciplinary healthcare team, and involving caregivers in any capacity. The intervention components found in this study can be leveraged in efforts to improve perioperative cancer care based on the resources available and the specific needs of the patients. In addition, different types of surgeons have different practices, workflows, and practice cultures. Interventions that work for one practice culture may need to be adapted to translate into other practice cultures. In conjunction with results of design focused, formative interviews with palliative care teams and surgeons, the evidence synthesized through this systematic review will be used to build an interdisciplinary teaming intervention [96]. Future interventions might prioritize expanding the roles of clinical team members to include educating patients, engaging the patient's caregiver, and ensuring sufficient follow-up towards improving quality for patients in the perioperative period.

## Supporting information

**S1 Checklist. PRISMA checklist.**
(DOCX)

**S1 File.**
(DOCX)

## Acknowledgments

This paper was presented at the Society of General Internal Medicine (SGIM) meeting in Orlando, Florida in April 2022 and the Academy of Management symposium presentation in Seattle, Washington in August 2022. The contents do not represent the views of the U.S. Department of Veterans Affairs or the U.S. Government.

## Author Contributions

**Conceptualization:** Bhagvat J. Maheta, Nainwant K. Singh, Karl A. Lorenz, Sydney M. Dy, Jonathan Bergman, John T. Leppert, Karleen F. Giannitrapani.

**Data curation:** Bhagvat J. Maheta, Nainwant K. Singh, Sarina Fereydooni, Karleen F. Giannitrapani.

**Formal analysis:** Bhagvat J. Maheta, Nainwant K. Singh, Karl A. Lorenz, Sydney M. Dy, Jonathan Bergman, John T. Leppert, Karleen F. Giannitrapani.

**Funding acquisition:** Karleen F. Giannitrapani.

**Investigation:** Bhagvat J. Maheta, Nainwant K. Singh, Karl A. Lorenz, Sydney M. Dy, Karleen F. Giannitrapani.

**Methodology:** Bhagvat J. Maheta, Nainwant K. Singh, Karl A. Lorenz, Sarina Fereydooni, Sydney M. Dy, Hong-nei Wong, Karleen F. Giannitrapani.

**Project administration:** Bhagvat J. Maheta, Karleen F. Giannitrapani.

**Resources:** Bhagvat J. Maheta, Karl A. Lorenz, Sarina Fereydooni, Sydney M. Dy, Hong-nei Wong, Jonathan Bergman, John T. Leppert, Karleen F. Giannitrapani.

**Software:** Bhagvat J. Maheta, Karl A. Lorenz, Sarina Fereydooni, Hong-nei Wong, Jonathan Bergman, John T. Leppert, Karleen F. Giannitrapani.

**Supervision:** Karl A. Lorenz, Sydney M. Dy, Jonathan Bergman, John T. Leppert, Karleen F. Giannitrapani.

**Validation:** Nainwant K. Singh, Karl A. Lorenz, Sarina Fereydooni, Karleen F. Giannitrapani.

**Visualization:** Bhagvat J. Maheta, Nainwant K. Singh, Karl A. Lorenz, Karleen F. Giannitrapani.

**Writing – original draft:** Bhagvat J. Maheta, Nainwant K. Singh, Karleen F. Giannitrapani.

**Writing – review & editing:** Bhagvat J. Maheta, Nainwant K. Singh, Karl A. Lorenz, Sarina Fereydooni, Sydney M. Dy, Hong-nei Wong, Jonathan Bergman, John T. Leppert, Karleen F. Giannitrapani.

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
