## [Decision Letter · Decision Letter 0]

31 Jul 2023

PONE-D-23-16022Interdisciplinary Interventions that Improve Patient-Reported Outcomes in Perioperative Cancer Care: A Systematic Review of Randomized Control TrialsPLOS ONE

Dear Dr. Giannitrapani,

Thank you for submitting your manuscript to PLOS ONE. After careful consideration, we feel that it has merit but does not fully meet PLOS ONE’s publication criteria as it currently stands. Therefore, we invite you to submit a revised version of the manuscript that addresses the points raised during the review process.

We look forward to receiving your revised manuscript.

Kind regards,

Sunil Shrestha

Academic Editor

PLOS ONE

“This project was presented at the Society of General Internal Medicine (SGIM) meeting in Orlando, Florida in April 2022. This project was also presented as part of the Academy of Management symposium presentation titled “Advancing Interdisciplinary Teamwork: Processes Supporting High-Quality and Innovative Performance” in Seattle, Washington in August 2022. Dr. Giannitrapani is supported by a VA Career Development Award (19-075). Dr. Nainwant Singh is supported by an Advanced Fellowship in Health Services Research sponsored by the Elizabeth Dole Center of Excellence and VA Center for Innovation to Implementation. The contents do not represent the views of the U.S. Department of Veterans Affairs or the U.S. Government.”

“KG is supported by a VA Career Development Award (19-075). The full name of the funder if the Veterans Affairs and this is the link to the funder website: https://www.research.va.gov/funding/cdp.cfm

4. Please include your tables as part of your main manuscript and remove the individual files. Please note that supplementary tables (should remain/ be uploaded) as separate "supporting information" files

Additional Editor Comments:

Comments from Editor and Reviewers.

Introduction

The introduction provides a comprehensive background on the importance of interdisciplinary teams in healthcare, particularly in the context of cancer care. It effectively highlights the dynamic nature of patient care and the need for various clinical team members to adjust their roles accordingly. The focus on the perioperative period is justified due to its critical role in cancer treatment and the potential for interdisciplinary collaboration to improve outcomes.

Clear Research Objectives: The introduction provides a general overview of the importance of interdisciplinary teams in perioperative cancer care. However, it would be beneficial to explicitly state the research objectives and research questions to provide readers with a clear understanding of what the systematic review aims to achieve. For example, stating specific research questions related to the structures and processes of interdisciplinary interventions and their impact on patient-reported outcomes would enhance the clarity of the study's purpose.

Literature Review: While the introduction touches on the significance of interdisciplinary teams in cancer care, it would be advantageous to include a more focused and in-depth review of existing literature related to interdisciplinary interventions in perioperative cancer care. Providing a summary of relevant prior studies and their findings would help contextualize the current review and highlight the research gaps being addressed.

Justification for Narrative Synthesis: The manuscript states that a narrative systematic review is conducted. However, there should be a justification for selecting this synthesis approach over a more quantitative meta-analysis. Discussing any anticipated heterogeneity among the included studies that might hinder meaningful statistical pooling would add credibility to the choice of narrative synthesis.

Focus on Patient-Reported Outcomes (PROs): The introduction mentions that the review aims to understand how clinical team members might improve perioperative care with a focus on patient-reported outcomes (PROs). It would be beneficial to clarify the specific PROs of interest and why they are relevant in assessing the effectiveness of interdisciplinary interventions. Additionally, discussing the advantages and limitations of relying solely on PROs in this context would add depth to the rationale.

Inclusion and Exclusion Criteria: The introduction briefly mentions the focus on randomized controlled trials (RCTs), but it does not explicitly outline the inclusion and exclusion criteria for study selection. Including a clear and concise statement of the criteria at the end of the introduction would be helpful for readers to understand the eligibility requirements for study inclusion.

Scope of Review: The introduction broadly mentions "structures and processes" in existing interdisciplinary interventions, but it could benefit from providing a more specific overview of the aspects of interventions that will be examined. Clearly defining the key elements (e.g., types of interventions, specific team member roles) to be considered during the review will guide readers in understanding the scope of the study.

Methods

Inclusion and Exclusion Criteria: The inclusion and exclusion criteria are clearly stated, but it would be helpful to provide a rationale or references to support some of the decisions, especially regarding the exclusion of certain interventions (art/music therapy, exercise therapy, diet changes, or spiritual therapy). Justifying these exclusions will enhance the transparency of the review process.

Data Abstraction: The data abstraction process is well-described, and it is commendable that conflicts were resolved through consensus discussions. However, it would be beneficial to report the inter-rater agreement (e.g., Cohen's kappa) between the two reviewers during both the title/abstract screening and full-text screening phases. Reporting inter-rater agreement helps assess the reliability of the review process.

Bias Assessment: The use of the Cochrane Risk of Bias tool for randomized controlled trials is appropriate. However, it would be valuable to include a table summarizing the risk of bias assessment for each included study to provide readers with a quick overview of the methodological quality of the evidence.

Data Synthesis: The narrative synthesis approach is justified due to the heterogeneity among the studies. However, to enhance the transparency and replicability of the synthesis process, consider providing a detailed description of how the narrative synthesis was conducted. For instance, describe how the themes were derived, how evidence was extracted and compared across studies, and the steps taken to ensure the accuracy and reliability of the synthesis.

Minimal Clinically Important Difference (MCID): The use of MCID to assess clinically meaningful improvement in PROs is reasonable. However, it would be beneficial to elaborate on the sources or references used to determine the MCID values for the different PRO measures. Additionally, if there were cases where MCID values were not available, clarify how the decision was made to rely solely on statistical significance.

Reporting of Results: The manuscript briefly mentions the findings of the narrative synthesis but does not provide specific details on the identified structures and processes in interdisciplinary interventions that improved perioperative PROs. Consider expanding the results section to include key findings and themes emerging from the synthesis, with supporting evidence from the included studies

Clarity in Reporting Results: The results section would benefit from a more organized and concise presentation of the findings. Consider dividing the results into subsections based on the key themes identified, such as "Common Roles," "Team Structures," "Team Processes," and "Caregiver Involvement." Presenting the results in this manner will make it easier for readers to grasp the main findings and their implications.

Elaboration on PRO Measures: While the manuscript mentions that 15 different PROs were reported across the included studies, it does not provide specific details about these measures or their psychometric properties. It would be helpful to include a brief description of the PRO measures used and their reliability and validity in measuring patient outcomes. This information would enhance the readers' understanding of the outcomes assessed in the studies.

Discussion: The discussion section should expand on the implications of the findings and their potential significance for clinical practice and research. Consider discussing how these findings align with existing literature on interdisciplinary team interventions in cancer care and whether any gaps or contradictions were observed. Additionally, discuss the limitations of the review and potential areas for future research to address those limitations.

Limitations: The manuscript briefly mentions the limitations of the study but could benefit from a more comprehensive and critical assessment of potential biases and sources of uncertainty. Address the potential limitations related to the search strategy, study selection, data synthesis, and risk of bias assessment. Acknowledge any heterogeneity across the included studies and discuss how this might have influenced the overall findings and interpretation.

Replicability and Transparency: To enhance the replicability of the review, consider providing a PRISMA flowchart detailing the screening and selection process for the included studies. This flowchart will offer readers a clear visual representation of the study selection process. Additionally, consider sharing the full abstraction form (Appendix S2) and risk of bias assessments (Appendix S3) in the manuscript or as supplementary material to enhance the transparency of the review process.

Interpretation of Results: The manuscript states that "no included studies had an overall high risk of bias," but it would be helpful to elaborate on the overall risk of bias assessment and the implications of potential bias on the review's findings. Provide a more detailed discussion on how the overall quality of evidence may influence the confidence in the conclusions drawn.

Recommendations and Practical Implications: In the conclusion section, consider offering specific recommendations for healthcare professionals and researchers based on the review's findings. Provide practical implications of the identified common roles, team structures, and team processes to improve perioperative cancer care and enhance patient outcomes.

Reviewer’s report:

The authors aim to characterize the structural and procedural components of interdisciplinary interventions in perioperative cancer care that have a positive impact in patient-reported outcomes. There is variability in the intervention used and interdisciplinary team members in interventional studies in perioperative cancer care showing improvement in patient-reported outcomes.

Overall, the study would seem to have constructive influence in practice as well as research.

As of my awareness following revisions should be made to the manuscripts.

• The term interdisciplinary intervention should be defined in the manuscript since there is a very thin line between definition of terms multidisciplinary and interdisciplinary in health research. Authors can go through a review entitled “Multidisciplinarity, interdisciplinarity and transdisciplinarity in health research, services, education and policy: 1. Definitions, objectives, and evidence of effectiveness” by Choi and Pak. This further seems necessary due to variability in the team processes in the included studies.

• Page no. 2: Abstract, Results, first sentence: patient-reported outcomes is appropriate in place of patient-reported quality outcomes.

• Page no. 4: The statement “Since the perioperative period of cancer care has an increased incidence of morbidity and mortality, it serves as an opportunity for leveraging diverse clinical team members to improve outcomes for patients” does not seem to be well supported by the cited literature. It would be better if the sentence is re-written or if appropriate literature supporting the statement be cited.

• Page no. 4: last paragraph, first sentence: systematic review would be appropriate be than narrative systematic review.

• Please go through manuscript again for eliminating typos, spelling, grammar, and phrasing errors.

Reviewer 2

Dear Authors,

Your article “Interdisciplinary Interventions that Improve Patient-Reported Outcomes in Perioperative Cancer Care: A Systematic Review of Randomized Control Trials” systematic review of randomized meets the criteria and requirement of PLOS One and delivers some notable information’s and context.

I find your writing mostly to be clear and concise and appropriately deliver the theme /importance of team while managing perioperative cancer patients. The paper properly highlights the importance of surgery, its complications and the interdisciplinary interventions such as education, engaging with patient caregivers, communication, longitudinal follow-up in perioperative period in cancer patients for better patient reported outcomes. I found the search strategy, PRISMA-P flow charts and all the relevant information attached within and found the article genuinely suitable for publication in this journal. However, I do have some minor comments regarding your paper. Please try to address these comments and suggestions.

1. Page 3 Para 2nd:

“Both cancer and its treatment touches and often impairs diverse aspects of quality of life (e.g., physical, emotional, existential, and social); because needs fluctuate over time, an interdisciplinary team can be leveraged to support patients’ changing needs so that the care is customized to each patient’s changing needs.”

Comments: The theme I got from this sentence is that “Cancer and its treatment impairs the quality of life of patients (physical, emotional, existential, and social) and hence interdisciplinary support and team can be established in order to support the patients’ changing needs.” However, this sentence while reading sounds a bit complex and mismatched and can be complex for the readers. Hence, I request you to please re-write the sentence without destroying the essence.

2. Page 7, Paragraph 2nd:

“The main outcomes collected were any form of patient-reported outcomes (PROs).” In the same paragraph where you’ve also written “Through different patient-reported outcomes……”. This sentence also lacks PROs abbreviation consistency.

Comments: You have already abbreviated patient-reported outcomes as PROs in Page 4, 4th Paragraph. Please maintain abbreviation consistency.

3. Page 12-13: “There are not enough included studies, and not enough information regarding team structures in the included studies to draw conclusions about the most effective team structures in improving PROs in perioperative cancer care.”

Comments: You mentioned how there are not enough studies and information regarding team structures to draw conclusion regarding PROs in perioperative cancer care. Regarding this statement you haven’t addressed the various reasons for such low number of studies in the discussion section. I feel this needs to be addressed since you have mentioned your aim to identify the team structure responsible for interdisciplinary interventions for better PROs.

4. Since you’ve mentioned the aim of the study to identify the team structure and process while exhibiting interdisciplinary interventions. Your conclusion includes the interventions that were responsible for improved PROs. However, it does not give the idea on the non-physician intervention clinical team member such as advanced practice providers (APP), including nurse practitioners and physician assistants, clinical nurse specialists, and registered nurses. Do any of these needs to be included in the conclusion section? Since you’ve mentioned how advanced practice providers in the intervention, regardless of the specific role they performed were associated with an improvement in PROs compared to the control group and the roles of nurse practitioners in cancer patients. I feel your conclusion can be more concrete if you mention both the idea of the non-physician clinical team members responsible and the interventions provided for better PROs. Regarding the limited papers of the inclusion of APP, you can go ahead with the similar information in your conclusion. You can also include the role of communication as an intervention in the conclusion section since you have addressed how communication has helped in improving PROs as an interventional role (Page 13, para 2nd)

5. Please look out for the minor typing errors, mismatched spelling and punctuation marks.

Thank you. I feel these comments are very simple and can be easily addressed. Best of luck!

Reviewers' comments:

Reviewer's Responses to Questions

**Comments to the Author**

1. Is the manuscript technically sound, and do the data support the conclusions?

Reviewer #1: Yes

Reviewer #2: Yes

2. Has the statistical analysis been performed appropriately and rigorously? 

Reviewer #1: N/A

Reviewer #2: Yes

3. Have the authors made all data underlying the findings in their manuscript fully available?

Reviewer #1: Yes

Reviewer #2: Yes

4. Is the manuscript presented in an intelligible fashion and written in standard English?

Reviewer #1: Yes

Reviewer #2: Yes

5. Review Comments to the Author

Reviewer #1: Title: Interdisciplinary Interventions that Improve Patient-Reported Outcomes in Perioperative Cancer Care: A Systematic Review of Randomized Control Trials

Version: 1 Date: 25 July 2023

Reviewer’s report:

The authors aim to characterize the structural and procedural components of interdisciplinary interventions in perioperative cancer care that have a positive impact in patient-reported outcomes. There is variability in the intervention used and interdisciplinary team members in interventional studies in perioperative cancer care showing improvement in patient-reported outcomes.

Overall, the study would seem to have constructive influence in practice as well as research.

As of my awareness following revisions should be made to the manuscripts.

• The term interdisciplinary intervention should be defined in the manuscript since there is a very thin line between definition of terms multidisciplinary and interdisciplinary in health research. Authors can go through a review entitled “Multidisciplinarity, interdisciplinarity and transdisciplinarity in health research, services, education and policy: 1. Definitions, objectives, and evidence of effectiveness” by Choi and Pak. This further seems necessary due to variability in the team processes in the included studies.

• Page no. 2: Abstract, Results, first sentence: patient-reported outcomes is appropriate in place of patient-reported quality outcomes.

• Page no. 4: The statement “Since the perioperative period of cancer care has an increased incidence of morbidity and mortality, it serves as an opportunity for leveraging diverse clinical team members to improve outcomes for patients” does not seem to be well supported by the cited literature. It would be better if the sentence is re-written or if appropriate literature supporting the statement be cited.

• Page no. 4: last paragraph, first sentence: systematic review would be appropriate be than narrative systematic review.

• Please go through manuscript again for eliminating typos, spelling, grammar, and phrasing errors.

Reviewer #2: Dear Authors,

Your article “Interdisciplinary Interventions that Improve Patient-Reported Outcomes in Perioperative Cancer Care: A Systematic Review of Randomized Control Trials” systematic review of randomized meets the criteria and requirement of PLOS One and delivers some notable information’s and context.

I find your writing mostly to be clear and concise and appropriately deliver the theme /importance of team while managing perioperative cancer patients. The paper properly highlights the importance of surgery, its complications and the interdisciplinary interventions such as education, engaging with patient caregivers, communication, longitudinal follow-up in perioperative period in cancer patients for better patient reported outcomes. I found the search strategy, PRISMA-P flow charts and all the relevant information attached within and found the article genuinely suitable for publication in this journal. However, I do have some minor comments regarding your paper. Please try to address these comments and suggestions.

1. Page 3 Para 2nd:

“Both cancer and its treatment touches and often impairs diverse aspects of quality of life (e.g., physical, emotional, existential, and social); because needs fluctuate over time, an interdisciplinary team can be leveraged to support patients’ changing needs so that the care is customized to each patient’s changing needs.”

Comments: The theme I got from this sentence is that “Cancer and its treatment impairs the quality of life of patients (physical, emotional, existential, and social) and hence interdisciplinary support and team can be established in order to support the patients’ changing needs.” However, this sentence while reading sounds a bit complex and mismatched and can be complex for the readers. Hence, I request you to please re-write the sentence without destroying the essence.

2. Page 7, Paragraph 2nd:

“The main outcomes collected were any form of patient-reported outcomes (PROs).” In the same paragraph where you’ve also written “Through different patient-reported outcomes……”. This sentence also lacks PROs abbreviation consistency.

Comments: You have already abbreviated patient-reported outcomes as PROs in Page 4, 4th Paragraph. Please maintain abbreviation consistency.

3. Page 12-13: “There are not enough included studies, and not enough information regarding team structures in the included studies to draw conclusions about the most effective team structures in improving PROs in perioperative cancer care.”

Comments: You mentioned how there are not enough studies and information regarding team structures to draw conclusion regarding PROs in perioperative cancer care. Regarding this statement you haven’t addressed the various reasons for such low number of studies in the discussion section. I feel this needs to be addressed since you have mentioned your aim to identify the team structure responsible for interdisciplinary interventions for better PROs.

4. Since you’ve mentioned the aim of the study to identify the team structure and process while exhibiting interdisciplinary interventions. Your conclusion includes the interventions that were responsible for improved PROs. However, it does not give the idea on the non-physician intervention clinical team member such as advanced practice providers (APP), including nurse practitioners and physician assistants, clinical nurse specialists, and registered nurses. Do any of these needs to be included in the conclusion section? Since you’ve mentioned how advanced practice providers in the intervention, regardless of the specific role they performed were associated with an improvement in PROs compared to the control group and the roles of nurse practitioners in cancer patients. I feel your conclusion can be more concrete if you mention both the idea of the non-physician clinical team members responsible and the interventions provided for better PROs. Regarding the limited papers of the inclusion of APP, you can go ahead with the similar information in your conclusion. You can also include the role of communication as an intervention in the conclusion section since you have addressed how communication has helped in improving PROs as an interventional role (Page 13, para 2nd)

5. Please look out for the minor typing errors, mismatched spelling and punctuation marks.

Thank you. I feel these comments are very simple and can be easily addressed. Best of luck!

6. PLOS authors have the option to publish the peer review history of their article (what does this mean?). If published, this will include your full peer review and any attached files.

Reviewer #1: No

Reviewer #2: No

---

## [Author Response · Author response to Decision Letter 0]

8 Sep 2023

Abstract

Reviewer Comments

Author Responses

Page no. 2: Abstract, Results, first sentence: patient-reported outcomes is appropriate in place of patient-reported quality outcomes. (Reviewer 1)

We have updated this sentence to say “patient-reported outcomes”

Introduction

Reviewer Comments

Author Responses

The introduction provides a comprehensive background on the importance of interdisciplinary teams in healthcare, particularly in the context of cancer care. It effectively highlights the dynamic nature of patient care and the need for various clinical team members to adjust their roles accordingly. The focus on the perioperative period is justified due to its critical role in cancer treatment and the potential for interdisciplinary collaboration to improve outcomes. (Reviewer 1)

Thank you for reviewing our manuscript.

Clear Research Objectives: The introduction provides a general overview of the importance of interdisciplinary teams in perioperative cancer care. However, it would be beneficial to explicitly state the research objectives and research questions to provide readers with a clear understanding of what the systematic review aims to achieve. For example, stating specific research questions related to the structures and processes of interdisciplinary interventions and their impact on patient-reported outcomes would enhance the clarity of the study's purpose. (Reviewer 1)

We have now clearly written the research objective and research questions at the end of the introduction:

“The purpose of this paper was to identify the structures (who is on the team and what are their roles) and processes (how the team functions and communicates) in existing interdisciplinary interventions that improved perioperative patient reported outcomes (PROs) in randomized controlled trials to garner insights on how future interventions might improve PROs in perioperative cancer care. We also aimed to understand the various components of interventions and the impact they had on PROs. This includes the types of interventions, specific team member roles, and caregiver involvement.”

Literature Review: While the introduction touches on the significance of interdisciplinary teams in cancer care, it would be advantageous to include a more focused and in-depth review of existing literature related to interdisciplinary interventions in perioperative cancer care. Providing a summary of relevant prior studies and their findings would help contextualize the current review and highlight the research gaps being addressed. (Reviewer 1)

We have summarized a few relevant articles to add context to the field of interdisciplinary interventions in perioperative cancer care:

“Recent interventions in perioperative cancer care have revolved around integrating interdisciplinary teams to improve patient education, follow-up, and rehabilitation, leading to improved patient outcomes.10 For example, in a randomized control trial where a nurse practitioner joined the oncology team to discuss hospice, living wills, and advanced directives with patients metastatic cancer, there was an improvement in the patients’ quality of life, overall physical, mental, and emotional wellbeing.11”

“Despite a recent surge in interventions integrating interdisciplinary teams in perioperative cancer care, individual components on interventions such as structures and processes have not been analyzed. “

Justification for Narrative Synthesis: The manuscript states that a narrative systematic review is conducted. However, there should be a justification for selecting this synthesis approach over a more quantitative meta-analysis. Discussing any anticipated heterogeneity among the included studies that might hinder meaningful statistical pooling would add credibility to the choice of narrative synthesis. (Reviewer 1)

We have added justification for the narrative synthesis approach with the data analysis section of the Methods:

“Each included study used different methods, and reported different outcomes, which hinders meaningful statistical pooling.”

Focus on Patient-Reported Outcomes (PROs): The introduction mentions that the review aims to understand how clinical team members might improve perioperative care with a focus on patient-reported outcomes (PROs). It would be beneficial to clarify the specific PROs of interest and why they are relevant in assessing the effectiveness of interdisciplinary interventions. Additionally, discussing the advantages and limitations of relying solely on PROs in this context would add depth to the rationale. (Reviewer 1)

In the Methods, we added:

“PROs of interest included the Quality of Life Scales (QLQ-c30), Hospital Anxiety and Depression Scales (HADS), Symptom Distress Scale (SDS), 12-Item Short Form Health Survey (SF-12), Functional Assessment of Cancer Therapy (FACT-C), Functional Assessment of Cancer Therapy (FACT-B), Self-rating Anxiety Scale (SAS), Self-rating Depression Scale (SDS), Profile of Mood States, Gracely Pain Scale, Center for Epidemiologic Studies Depression Scale, Stoma Self-Efficacy Scale, Numerical Rating Scale for pain intensity, American Pain Society- patient questionnaire, and the Mood Adjective Check List (MACL). These metrics help us better understand the impact of interventions on a person’s life.”

“PROs in the perioperative period can be used to quantitatively assess a patient's quality of life.29 Healthcare teams can utilize PROs as tools to improve patient outcomes, inform best practices, and refine patient care.30 Through different patient-reported outcomes (such as the Functional Assessment of Cancer Therapy Scale, QLQ-c30, Profile of Mood States, MACL, HADS), researchers can understand how study interventions may impact patient quality of life.29”

In the Discussion, we include limitations of relying on PROs:

“Further, PROs hold the bias that patients may have their own individual reference points for answering the questions.”

Inclusion and Exclusion Criteria: The introduction briefly mentions the focus on randomized controlled trials (RCTs), but it does not explicitly outline the inclusion and exclusion criteria for study selection. Including a clear and concise statement of the criteria at the end of the introduction would be helpful for readers to understand the eligibility requirements for study inclusion. (Reviewer 1)

Thank you for this comment. We have explicitly outlined the inclusion and exclusion criteria for study selection in the second paragraph of the Methods: 

“Inclusion criteria were: RCTs, adults (18 and over), English only (for feasibility), PROs, inclusion of clinical team members as part of the healthcare team, perioperative period (defined as 30 days before and up to 90 days after surgery), and cancer care in the in-patient or out-patient/ ambulatory setting.15 Exclusion criteria were: non-RCTs, protocol papers, pediatric population, studies not written in English, no PROs, no clinical team member in the intervention, not during the perioperative period, no patients with cancer, or interventions testing the technical execution of surgery, radiation, pharmaceutical delivery, or other clinical therapies.”

Scope of Review: The introduction broadly mentions "structures and processes" in existing interdisciplinary interventions, but it could benefit from providing a more specific overview of the aspects of interventions that will be examined. Clearly defining the key elements (e.g., types of interventions, specific team member roles) to be considered during the review will guide readers in understanding the scope of the study. (Reviewer 1)

We have clearly defined structures and processes as well as the components of interventions at the end of the Introduction:

“The purpose of this paper was to identify the structures (who is on the team and what are their roles) and processes (how the team functions and communicates) in existing interdisciplinary interventions that improved perioperative patient reported outcomes (PROs) in randomized controlled trials to garner insights on how future interventions might improve PROs in perioperative cancer care. We also aimed to understand the various components of interventions and the impact they had on PROs. This includes the types of interventions, specific team member roles, and caregiver involvement.”

The term interdisciplinary intervention should be defined in the manuscript since there is a very thin line between definition of terms multidisciplinary and interdisciplinary in health research. Authors can go through a review entitled “Multidisciplinarity, interdisciplinarity and transdisciplinarity in health research, services, education and policy: 1. Definitions, objectives, and evidence of effectiveness” by Choi and Pak. This further seems necessary due to variability in the team processes in the included studies. (Reviewer 1)

Thank you for bringing up this point and suggesting a helpful reference. We have now defined interdisciplinary teams within the introduction as:

“Interdisciplinary teams can be defined as a team that synthesizes and harmonizes links between each individual discipline into a coordinated and coherent whole.8”

Page no. 4: The statement “Since the perioperative period of cancer care has an increased incidence of morbidity and mortality, it serves as an opportunity for leveraging diverse clinical team members to improve outcomes for patients” does not seem to be well supported by the cited literature. It would be better if the sentence is re-written or if appropriate literature supporting the statement be cited. (Reviewer 1)

This sentence has been re-written to:

“Since there is an increased incidence of morbidity and mortality in the perioperative period of cancer care, there is an opportunity for leveraging diverse clinical team members to improve outcomes for patients within this time period.10, 18-20”

The references for this sentence have also been updated to better support the claim being made that there is an opportunity for leveraging diverse clinical team members within this perioperative period of cancer care to improve outcomes for patients.

Page no. 4: last paragraph, first sentence: systematic review would be appropriate be than narrative systematic review. (Reviewer 1)

We have adjusted this sentence:

“Through this systematic review…”

Page 3 Para 2nd:

“Both cancer and its treatment touches and often impairs diverse aspects of quality of life (e.g., physical, emotional, existential, and social); because needs fluctuate over time, an interdisciplinary team can be leveraged to support patients’ changing needs so that the care is customized to each patient’s changing needs.”

Comments: The theme I got from this sentence is that “Cancer and its treatment impairs the quality of life of patients (physical, emotional, existential, and social) and hence interdisciplinary support and team can be established in order to support the patients’ changing needs.” However, this sentence while reading sounds a bit complex and mismatched and can be complex for the readers. Hence, I request you to please re-write the sentence without destroying the essence. (Reviewer 2)

We have adjusted the sentence to read:

“Both cancer and its treatment often impairs aspects of quality of life (e.g. physical, emotional, existential, and social); because patient needs fluctuate over time, an interdisciplinary team can be leveraged to support patients’ changing needs so that the care is customized for each patient.5,10”

Methods

Reviewer Comments

Author Responses

Inclusion and Exclusion Criteria: The inclusion and exclusion criteria are clearly stated, but it would be helpful to provide a rationale or references to support some of the decisions, especially regarding the exclusion of certain interventions (art/music therapy, exercise therapy, diet changes, or spiritual therapy). Justifying these exclusions will enhance the transparency of the review process. (Reviewer 1)

We have included a rationale within the Methods section:

“We also excluded studies of art/music therapy, exercise therapy, diet changes, or spiritual therapy for patients because these interventions have been extensively studied in the literature as effective programs on their own. These represent specific clinical interventions that have been shown to be effective on their own, regardless of what team members facilitating the intervention, and we wanted this systematic review to focus on the components of interventions (e.g. team structure and processes) that impacted PROs.24–26”

Data Abstraction: The data abstraction process is well-described, and it is commendable that conflicts were resolved through consensus discussions. However, it would be beneficial to report the inter-rater agreement (e.g., Cohen's kappa) between the two reviewers during both the title/abstract screening and full-text screening phases. Reporting inter-rater agreement helps assess the reliability of the review process. (Reviewer 1)

We included the Cohen’s Kappa values for each screening phase in the methods.

Title/ abstract screening: (Cohen’s Kappa = 0.94022)

Full text screening: (Cohen’s Kappa = 0.84632)

Bias Assessment: The use of the Cochrane Risk of Bias tool for randomized controlled trials is appropriate. However, it would be valuable to include a table summarizing the risk of bias assessment for each included study to provide readers with a quick overview of the methodological quality of the evidence. (Reviewer 1)

The Risk of Bias for each included article is included in Appendix S3.

Data Synthesis: The narrative synthesis approach is justified due to the heterogeneity among the studies. However, to enhance the transparency and replicability of the synthesis process, consider providing a detailed description of how the narrative synthesis was conducted. For instance, describe how the themes were derived, how evidence was extracted and compared across studies, and the steps taken to ensure the accuracy and reliability of the synthesis. (Reviewer 1)

We have described the process of the narrative synthesis within the Methods section:

“The entire intervention description provided was compared to the other intervention descriptions to find similar components across the various interventions. We analyzed the expanded roles of clinical team members and additional components of interventions that were done by the design of the intervention on top of the usual care provided to the control group for the respective article. For articles where the control group received usual care and additional care such as physical therapy sessions, we considered the roles and intervention components that were added on top of the control. Different clinical team member roles, as well as intervention components, were then categorized based on if they showed a clinically meaningful improvement in PROs compared to the control group in their respective studies. We used this to extrapolate results and recommendations to improve perioperative cancer care from these sparse studies. To ensure the accuracy and reliability of the synthesis, all intervention components extrapolated were discussed and modified through an interactive process with the entire team.”

Minimal Clinically Important Difference (MCID): The use of MCID to assess clinically meaningful improvement in PROs is reasonable. However, it would be beneficial to elaborate on the sources or references used to determine the MCID values for the different PRO measures. Additionally, if there were cases where MCID values were not available, clarify how the decision was made to rely solely on statistical significance. (Reviewer 1)

In cases where MCID values were not available, we relied on statistical significance since it was the only reliable metric available. We clarified this in the Methods:

“In studies with PROs that as of yet do not have established MCID, we only considered statistical significance since this was the only reliable metric available to analyze the data from the included studies.”

The sources that determine the MCID values for each PRO are provided in Appendix S5:

“Appendix S5 also provides the ranges for the included PROs that determine MCID based on the most recent published data available.”

Page 7, Paragraph 2nd:

“The main outcomes collected were any form of patient-reported outcomes (PROs).” In the same paragraph where you’ve also written “Through different patient-reported outcomes……”. This sentence also lacks PROs abbreviation consistency.

Comments: You have already abbreviated patient-reported outcomes as PROs in Page 4, 4th Paragraph. Please maintain abbreviation consistency. (Reviewer 2)

We have adjusted this sentence as suggested:

“The main outcomes collected were any form of PROs.”

Results

Reviewer Comments

Author Responses

Reporting of Results: The manuscript briefly mentions the findings of the narrative synthesis but does not provide specific details on the identified structures and processes in interdisciplinary interventions that improved perioperative PROs. Consider expanding the results section to include key findings and themes emerging from the synthesis, with supporting evidence from the included studies. (Reviewer 1)

We have included this in the Results section:

“Team structures varied vastly across the interventions, wrangling from a nurse and physician team,40,45,50,54,56,57 a nurse and another member of the interdisciplinary healthcare team,43,52,60,66 and teams including three or more interdisciplinary healthcare team members.41,42,44,47-49,55,61,63,64,71 Team processes included consistent communication within the interdisciplinary healthcare team at multiple different points in time.40,44,45,51,56,62,64,67,71 Intervention components included clinical team members roles of group education, patient/caregiver education, and clinical follow-up, contact with the patient throughout the intervention, and caregiver involvement.”

Clarity in Reporting Results: The results section would benefit from a more organized and concise presentation of the findings. Consider dividing the results into subsections based on the key themes identified, such as "Common Roles," "Team Structures," "Team Processes," and "Caregiver Involvement." Presenting the results in this manner will make it easier for readers to grasp the main findings and their implications. (Reviewer 1)

We have included the following subheadings to allow for greater organization within the Results section:

Literature Selection

Common Roles

Common Roles of Advanced Practice Providers

Common Roles of Nursing Team Members

Contact with Patient Throughout Intervention

Team Structures

Team Processes

Caregiver Involvement

Elaboration on PRO Measures: While the manuscript mentions that 15 different PROs were reported across the included studies, it does not provide specific details about these measures or their psychometric properties. It would be helpful to include a brief description of the PRO measures used and their reliability and validity in measuring patient outcomes. This information would enhance the readers' understanding of the outcomes assessed in the studies. (Reviewer 1)

The individual PROs, and their reported scales and values for MCID based on the current literature has been included in Appendix S5

Discussion

Reviewer Comments

Author Responses

Discussion: The discussion section should expand on the implications of the findings and their potential significance for clinical practice and research. Consider discussing how these findings align with existing literature on interdisciplinary team interventions in cancer care and whether any gaps or contradictions were observed. Additionally, discuss the limitations of the review and potential areas for future research to address those limitations. (Reviewer 1)

We have included how our findings align with the existing literature within the Discussion. Specifically, we reference other studies that have utilized intervention components that were found to be effective in this review as well as demonstrate the practical utilization of this knowledge by sharing the care in paragraph 4 of the Discussion:

“ Inclusion of advanced practice providers has been shown to reduce overall cost and provide support for physicians.82 Given the well-known workforce constraints for palliative care in cancer, advanced practice providers are an ideal alternative to maintain quality of care while considering overall healthcare expenditure.83 Oncology advanced practice providers can lead follow-up visits and educational sessions for patients in addition to administrative and clinical responsibilities, making this a viable opportunity specifically in the context of perioperative cancer care.84”

“ In the literature, several studies have shown that using consistent communication within the healthcare team can be an effective part of interventions for treating patients with different types of cancer in different stages of treatment.5,88,89 Longitudinal follow-up and effective communication among the healthcare team, with referrals to healthcare team members, may be used to improve the quality of life for patients with cancer in the perioperative period.90”

“Studies focusing on cancer care have shown the effectiveness of involving caregivers to allow for shared decision making to improve mental health and resilience, which is consistent with improvements in the QLQ-c30, Stoma Self-Efficacy Scale, HADS, and MACL PROs.92,93 Including caregivers in cancer interventions has also translated to improved patient satisfaction, suggesting it may be a worthwhile component to include in the treatment of patients with cancer in the perioperative period.94”

Limitations and future directions are included in the 7th paragraph of the Discussion.

Limitations: The manuscript briefly mentions the limitations of the study but could benefit from a more comprehensive and critical assessment of potential biases and sources of uncertainty. Address the potential limitations related to the search strategy, study selection, data synthesis, and risk of bias assessment. Acknowledge any heterogeneity across the included studies and discuss how this might have influenced the overall findings and interpretation. (Reviewer 1)

Limitations have been included within the discussion:

“Our study can be considered in light of the following limitations. The incorporation of advanced practice providers in the interdisciplinary healthcare team is recent and only three articles that had an intervention with advanced practice providers were included from a limited body of literature. We anticipate an increasing number of publications with interventions that include advanced practice providers. The patient reported outcomes included were heterogeneous and thus a meta-analysis or further synthesis is not possible. Further, PROs hold the bias that patients may have their own individual reference points for answering the questions.95 Our abstraction form did not capture the type of surgeon, which may be relevant in understanding PROs in different perioperative contexts. Furthermore, the search strategy itself, and the screening process, may have missed certain articles. The included studies were heterogeneous in nature, which did not allow for statistical analysis, thus limiting the conclusions formed in this study. Finally, the included studies did not provide sufficient information in the manuscript regarding the purpose of surgeon interaction with the patient during the perioperative period (communicate the purpose and expect results of surgery, identify patients who are appropriate for surgery, optimize patients for surgery to minimize risks, or identify and manage complications postoperatively), which may have added an additional layer of understanding.”

Replicability and Transparency: To enhance the replicability of the review, consider providing a PRISMA flowchart detailing the screening and selection process for the included studies. This flowchart will offer readers a clear visual representation of the study selection process. Additionally, consider sharing the full abstraction form (Appendix S2) and risk of bias assessments (Appendix S3) in the manuscript or as supplementary material to enhance the transparency of the review process. (Reviewer 1)

The PRISMA flowchart has been included as Figure 1 and the full abstraction form and risk of bias assessments have been included in Appendix S2 and S3 respectively.

Interpretation of Results: The manuscript states that "no included studies had an overall high risk of bias," but it would be helpful to elaborate on the overall risk of bias assessment and the implications of potential bias on the review's findings. Provide a more detailed discussion on how the overall quality of evidence may influence the confidence in the conclusions drawn. (Reviewer 1)

In the discussion, we added:

“Since no included articles had an overall high risk of bias, all studies were of high quality and thus minimize potential bias in the conclusions drawn from this systematic review.”

Page 12-13: “There are not enough included studies, and not enough information regarding team structures in the included studies to draw conclusions about the most effective team structures in improving PROs in perioperative cancer care.”

Comments: You mentioned how there are not enough studies and information regarding team structures to draw conclusion regarding PROs in perioperative cancer care. Regarding this statement you haven’t addressed the various reasons for such low number of studies in the discussion section. I feel this needs to be addressed since you have mentioned your aim to identify the team structure responsible for interdisciplinary interventions for better PROs. (Reviewer 2)

We added the following sentence explaining how we think there are limited number of studies since the incorporation of extended team members is a relatively recent aspect of healthcare delivery within the Discussion section:

“The incorporation of advanced practice providers in the interdisciplinary healthcare team is recent and only three articles that had an intervention with advanced practice providers were included from a limited body of literature. We anticipate an increasing number of publications with interventions that include advanced practice providers.”

Conclusion

Reviewer Comments

Author Responses

Recommendations and Practical Implications: In the conclusion section, consider offering specific recommendations for healthcare professionals and researchers based on the review's findings. Provide practical implications of the identified common roles, team structures, and team processes to improve perioperative cancer care and enhance patient outcomes. (Reviewer 1)

We have included a specific recommendation based on the review findings:

“The intervention components found in this study can be leveraged in efforts to improve 

perioperative cancer care based on the resources available and the specific needs of the patients.”

Since you’ve mentioned the aim of the study to identify the team structure and process while exhibiting interdisciplinary interventions. Your conclusion includes the interventions that were responsible for improved PROs. However, it does not give the idea on the non-physician intervention clinical team member such as advanced practice providers (APP), including nurse practitioners and physician assistants, clinical nurse specialists, and registered nurses. Do any of these needs to be included in the conclusion section? Since you’ve mentioned how advanced practice providers in the intervention, regardless of the specific role they performed were associated with an improvement in PROs compared to the control group and the roles of nurse practitioners in cancer patients. I feel your conclusion can be more concrete if you mention both the idea of the non-physician clinical team members responsible and the interventions provided for better PROs. Regarding the limited papers of the inclusion of APP, you can go ahead with the similar information in your conclusion. You can also include the role of communication as an intervention in the conclusion section since you have addressed how communication has helped in improving PROs as an interventional role (Page 13, para 2nd) (Reviewer 2)

We clarified that any clinical team member (advanced practice providers, nurse practitioners, or registered nurses) can be utilized for specific roles within interventions, and studies have shown improved PROs regardless of provider type. We also mentioned how communication can be utilized to improve PROs:

“These included expanding the roles of any clinical team members (advanced practice providers, nurse practitioners, or registered nurses) to include either group education, patient/caregiver education or clinical follow-up since they all demonstrated improved PROs. Other intervention components included longitudinal follow-up for longer than six months, increasing consistent communication among the interdisciplinary healthcare team, and involving caregivers in any capacity.”

Overall

Reviewer Comments

Author Responses

We have ensured that we have adhered to all of PLOS ONE’s style requirements

Thank you for stating the following in the Acknowledgments Section of your manuscript:

“This project was presented at the Society of General Internal Medicine (SGIM) meeting in Orlando, Florida in April 2022. This project was also presented as part of the Academy of Management symposium presentation titled “Advancing Interdisciplinary Teamwork: Processes Supporting High-Quality and Innovative Performance” in Seattle, Washington in August 2022. Dr. Giannitrapani is supported by a VA Career Development Award (19-075). Dr. Nainwant Singh is supported by an Advanced Fellowship in Health Services Research sponsored by the Elizabeth Dole Center of Excellence and VA Center for Innovation to Implementation. The contents do not represent the views of the U.S. Department of Veterans Affairs or the U.S. Government.”

“KG is supported by a VA Career Development Award (19-075). The full name of the funder if the Veterans Affairs and this is the link to the funder website: https://www.research.va.gov/funding/cdp.cfm

Please include your amended statements within your cover letter; we will change the online submission form on your behalf. (Editor)

We have removed any funding-related text from the manuscript. 

We have also included the updated funding statement in the Cover Letter:

“Dr. Karleen Giannitrapani is supported by a VA Career Development Award (19-075). The full name of the funder is the Veterans Affairs and this is the link to the funder website: https://www.research.va.gov/funding/cdp.cfm.

Dr. Nainwant Singh is supported by an Advanced Fellowship in Health Services Research sponsored by the Elizabeth Dole Center of Excellence and VA Center for Innovation to Implementation. The full name of the funder is the Veterans Affairs and this is the link to the funder website: https://www.hsrd.research.va.gov/centers/dole/

We note that you have stated that you will provide repository information for your data at acceptance. Should your manuscript be accepted for publication, we will hold it until you provide the relevant accession numbers or DOIs necessary to access your data. If you wish to make changes to your Data Availability statement, please describe these changes in your cover letter and we will update your Data Availability statement to reflect the information you provide. (Editor)

We are able and willing to provide all of our data, and are happy to keep our Data Availability statement as is. The data for this systematic review comes from 34 published randomized controlled trials. Their doi’s are listed here:

https://doi.org/10.7257/1053-816X.2012.31.2.83

https://doi.org/10.1016/j.ejon.2003.12.005

https://doi.org/10.1016/j.ijnurstu.2016.09.009

https://doi.org/10.1002/pon.1365

https://doi.org/10.1007/s00520-021-06072-5

https://doi.org/10.1038/sj.bjc.6604811

https://doi.org/10.1080/0284186X.2017.1358462

https://doi.org/10.1007/s00520-017-3981-4

https://doi.org/10.1016/j.ejon.2015.09.009

https://doi.org/10.1007/DCR.0b013e31820bc152

https://doi.org/10.1016/j.ijnurstu.2020.103565

https://doi.org/10.36468/pharmaceutical-sciences.spl.167

Juping Zheng, & Jiang, Y. (2019). The improvements of rapid recovery nursing model on perioperative indicators after surgery and its effect on unhealthy emotion in the clinical nursing of patients with lung cancer. Int J Clin Exp Med, 12(11), 12928–12935. DOI: IJCEM0100548

https://doi.org/10.1200/JCO.2012.48.1036

https://doi.org/10.1080/09515078808251418

https://doi.org/10.1007/s11136-011-0065-7

https://doi.org/10.1016/j.ejon.2019.02.006

https://doi.org/10.1016/j.ejon.2019.101707

https://doi.org/10.1097/SGA.0000000000000290

https://doi.org/10.1002/pon.899

https://doi.org/10.1016/j.lungcan.2018.10.025

https://doi.org/10.1200/JCO.2005.05.193

Francke, A. L., Garssen, B., Luiken, J. B., De Schepper, A. M. E., Grypdonck, M., & Abu-Saad, H. H. (1997). Effects of a nursing pain programme on patient outcomes. Psycho-Oncology, 6(4), 302–310. DOI: 10.1002/(SICI)1099-1611(199712)6:4<302::AID-PON284>3.0.CO;2-D

https://doi.org/10.1188/07.ONF.133-141

https://doi.org/10.1093/dote/doz076

Bahrami, M., Dehgani, S., Eghbali, M., & Daryabeigi, R. (2012). The effect of a care program on pain intensity of cancer patients who underwent surgery and hospitalized in Sayyed-Al-Shohada Hospital of Isfahan University of Medical Sciences in 2011. Iranian Journal of Nursing and Midwifery Research, 17(6), 408–413. PMID: 23922580

https://doi.org/10.1177/2057158516679790

https://doi.org/10.1186/s12912-022-01054-2

https://doi.org/10.1155/2022/9000331

https://doi.org/10.1155/2022/1035971

https://doi.org/10.1097/MD.0000000000026736

https://doi.org/10.1001/jamaoto.2021.3765

https://doi.org/10.1016/j.apjon.2022.02.007

https://doi.org/10.1155/2022/6862463

Please include your tables as part of your main manuscript and remove the individual files. Please note that supplementary tables (should remain/ be uploaded) as separate "supporting information" files. (Editor)

We have included the tables as part of the manuscript and have kept supplementary tables as “supporting information” files.

Please include captions for your Supporting Information files at the end of your manuscript, and update any in-text citations to match accordingly. Please see our Supporting Information guidelines for more information: http://journals.plos.org/plosone/s/supporting-information. (Editor)

We have added captions for the Supporting Information files at the end of the manuscript:

Supporting Information

Appendix S1: Search Strategy

Appendix S2: Abstraction Guide

Appendix S3: Risk of Bias of Included Studies

Appendix S4: Intervention Purpose and Results of 27 Included Studies

Appendix S5: Patient-Reported Quality Outcomes Reported

The authors aim to characterize the structural and procedural components of interdisciplinary interventions in perioperative cancer care that have a positive impact in patient-reported outcomes. There is variability in the intervention used and interdisciplinary team members in interventional studies in perioperative cancer care showing improvement in patient-reported outcomes.

Overall, the study would seem to have constructive influence in practice as well as research. (Reviewer 1)

Thank you for reviewing our paper.

Please go through manuscript again for eliminating typos, spelling, grammar, and phrasing errors. (Reviewer 1)

Please look out for the minor typing errors, mismatched spelling and punctuation marks. (Reviewer 2)

We have gone through the manuscript and updated all typos, spelling issues, grammar errors, and punctuation discrepancies.

Your article “Interdisciplinary Interventions that Improve Patient-Reported Outcomes in Perioperative Cancer Care: A Systematic Review of Randomized Control Trials” systematic review of randomized meets the criteria and requirement of PLOS One and delivers some notable information’s and context. (Reviewer 2)

Thank you for reviewing our paper.

I find your writing mostly to be clear and concise and appropriately deliver the theme /importance of team while managing perioperative cancer patients. The paper properly highlights the importance of surgery, its complications and the interdisciplinary interventions such as education, engaging with patient caregivers, communication, longitudinal follow-up in perioperative period in cancer patients for better patient reported outcomes. I found the search strategy, PRISMA-P flow charts and all the relevant information attached within and found the article genuinely suitable for publication in this journal. (Reviewer 2)

Thank you for reviewing our paper.

---

## [Editor Report · Decision Letter 1]

20 Sep 2023

PONE-D-23-16022R1Interdisciplinary Interventions that Improve Patient-Reported Outcomes in Perioperative Cancer Care: A Systematic Review of Randomized Control TrialsPLOS ONE

Dear Dr. Giannitrapani,

Thank you for submitting your manuscript to PLOS ONE. After careful consideration, we feel that it has merit but does not fully meet PLOS ONE’s publication criteria as it currently stands. Therefore, we invite you to submit a revised version of the manuscript that addresses the points raised during the review process.

We look forward to receiving your revised manuscript.

Kind regards,

Sunil Shrestha

Academic Editor

PLOS ONE

Journal Requirements:

Additional Editor Comments:

The authors have addressed the majority of the reviewers' comments. To facilitate the manuscript's acceptance, I recommend addressing some of the remaining minor comments.

Introduction:

-Specify research objectives and gaps in the literature.

-Provide citations to support claims.

-Improve transition and flow between sections.

-Offer more specific examples during discussions.

-Support statistics with references.

-Condense background information.

-Clarify terminology.

-Explicitly connect the systematic review to the introduction.

-State hypotheses or research questions.

-Emphasize the significance of the systematic review.

Methods:

-Structure the section with subsections.

-Summarize key components of the search strategy.

-Provide rationales for inclusion and exclusion criteria.

-Explain the rationale behind excluding certain interventions.

-Mention inter-rater agreement during data collection.

-Clarify criteria for determining "clinically meaningful" differences.

-Define and distinguish "teaming" from traditional teamwork.

-Explain the significance of family involvement in decision-making.

-Ensure accessibility and citation of appendices and supplementary materials.

Discussion:

-Discuss quality assessment criteria or tools.

-Analyze potential limitations and biases in included studies.

-Provide specific examples or statistics from reviewed studies.

-Include a section on clinical implications of the findings.

Conclusion:

-Specify practical applications and expected benefits of the intervention.

-Elaborate on the importance of factors like surgeon type and purpose of interaction.

-Summarize limitations of the systematic review for transparency.

---

## [Author Response · Author response to Decision Letter 1]

1 Nov 2023

The authors have addressed the majority of the reviewers' comments. To facilitate the manuscript's acceptance, I recommend addressing some of the remaining minor comments.

Thank you for taking the time to review our manuscript. We have addressed the remaining minor comments below.

Introduction:

Specify research objectives and gaps in the literature.

We have made the appropriate changes in the manuscript:

The gaps in literature are addressed in this sentence: “Despite a recent surge in interventions integrating interdisciplinary teams in perioperative cancer care, individual components of interventions, such as specific structures and processes have not been analyzed.”

The research objectives are addressed: “The purpose of this paper was to identify the structures (who is on the team and what are their roles) and processes (how the team functions and communicates) in existing interdisciplinary interventions that improved perioperative patient reported outcomes (PROs) in randomized controlled trials to garner insights on how future interventions might improve PROs in perioperative cancer care. We also aimed to understand the various components of interventions and the impact they had on PROs. This includes the types of interventions, specific team member roles, and caregiver involvement.”

Provide citations to support claims.

All claims have been double checked to ensure they are supported with citations.

Improve transition and flow between sections.

The last sentence of the first paragraph was updated to better tie into the second paragraph about quality of life for cancer care: “Leveraging non-physician clinical team members (e.g. advanced practice providers, clinical nurse specialists, and registered nurses) to participate on interdisciplinary teams can improve the quality of life for patients, especially in the context of cancer care.”

The first sentence of the third paragraph builds on the second paragraph by specifically providing one time point (the perioperative period) where teams can play a role in improving patient care: “The perioperative period is an exemplary context in which teams may play an important role in fostering better outcomes in cancer care.”

The last sentence of the fourth paragraph helps set up the gap in literature, which leads into the fifth paragraph: “The perioperative period is an exemplary context in which teams may play an important role in fostering better outcomes in cancer care.”

Offer more specific examples during discussions.

We have included specific examples within the Introduction to better support the points made: “ For example, in a randomized control trial where a nurse practitioner joined the oncology team to discuss hospice, living wills, and advanced directives with patients metastatic cancer, there was an improvement in the patients’ quality of life, overall physical, mental, and emotional wellbeing.”

Support statistics with references.

All statistics have been double checked to ensure they are supported with citations.

Condense background information.

We have removed unnecessary information in the Introduction where appropriate to condense this section.

Clarify terminology.

We have clarified all the terms used within the Introduction:

“A team can be either bounded, where members have static, defined roles, or dynamic, where members adjust roles continuously based on the task.”

“Leveraging non-physician clinical team members (e.g. advanced practice providers, clinical nurse specialists, and registered nurses) to participate on interdisciplinary teams can improve the quality of life for patients, especially in the context of cancer care”

“During the perioperative period, surgeons and surgical teams have many complex tasks which require interdisciplinary collaboration: (1) communicate the purpose and expect results of surgery (which may be curative, life-extending, or palliative), (2) identify patients who are appropriate for surgery based on relative risk of frailty or life expectancy, (3) optimize patients for surgery to minimize risks, and (4) quickly identify and manage complications postoperatively.”

Explicitly connect the systematic review to the introduction.

We have adjusted the Introduction’s last paragraph to explicitly connect the systematic review to all the points made in the introduction:

“The purpose of this paper was to identify the structures (who is on the team and what are their roles) and processes (how the team functions and communicates) in existing interdisciplinary interventions that improved perioperative patient reported outcomes (PROs) among patients with cancer in randomized controlled trials to garner insights on how future interventions might improve PROs in perioperative cancer care.”

State hypotheses or research questions.

We have included the research questions in the Introduction:

“The purpose of this paper was to identify the structures (who is on the team and what are their roles) and processes (how the team functions and communicates) in existing interdisciplinary interventions that improved perioperative patient reported outcomes (PROs) in randomized controlled trials to garner insights on how future interventions might improve PROs in perioperative cancer care. We also aimed to understand the various components of interventions and the impact they had on PROs to guide future interventions directed towards improving PROs in perioperative cancer care. This includes the types of interventions, specific team member roles, and caregiver involvement.”

Emphasize the significance of the systematic review.

We have emphasized the significance of the systematic review within the Introduction:

“Through this systematic review, we sought to understand how clinical team members might improve perioperative care and how their roles might be further modified or extended to improve perioperative care quality”

“The purpose of this paper was to identify the structures (who is on the team and what are their roles) and processes (how the team functions and communicates) in existing interdisciplinary interventions that improved perioperative patient reported outcomes (PROs) among patients with cancer in randomized controlled trials to garner insights on how future interventions might improve PROs in perioperative cancer care.”

Methods:

Structure the section with subsections.

We have included the following subsections within the Methods:

Study Selection, Data Collection, and Data Synthesis

Summarize key components of the search strategy.

The key components utilized in the search strategy, as well as the entire search strategy itself, are included in the Methods:

“Key concepts included in the search strategy included randomized control trials, cancer care, and clinical team members as shown in Appendix S1.”

Provide rationales for inclusion and exclusion criteria.

We have included a detailed explanation of why certain inclusion/ exclusion criteria were included:

“We also excluded studies of art/music therapy, exercise therapy, diet changes, or spiritual therapy for patients because these interventions have been extensively studied in the literature as effective programs on their own. These represent specific clinical interventions that have been shown to be effective on their own, regardless of what team members facilitating the intervention, and we wanted this systematic review to focus on the components of interventions (e.g. team structure and processes) that impacted PROs.”

Further rationales are also included in Table 1

Explain the rationale behind excluding certain interventions.

Specific interventions such as art/music therapy, exercise therapy, diet changes, or spiritual therapy were excluded in this study. Detailed rationale behind these exclusions is provided in the Methods:

““We also excluded studies of art/music therapy, exercise therapy, diet changes, or spiritual therapy for patients because these interventions have been extensively studied in the literature as effective programs on their own. These represent specific clinical interventions that have been shown to be effective on their own, regardless of what team members facilitating the intervention, and we wanted this systematic review to focus on the components of interventions (e.g. team structure and processes) that impacted PROs.”

Mention inter-rater agreement during data collection.

We have included the numeric inter-rater reliability for each stage of screening:

“During the title/abstract screening (inter-rater reliability; Cohen’s Kappa = 0.94022), we resolved all conflicts by a group consensus or by a “gold standard” reviewer (KG). We screened the full texts that remained after the title/abstract screening in a similar way. During the full-text screening (inter-rater reliability; Cohen’s Kappa = 0.84632), the reason for exclusion was also identified.”

Clarify criteria for determining "clinically meaningful" differences.

We have defined MCID and have also included Appendix S5 that helps determines “clinically meaningful” differences:

“MCID is defined as the smallest possible change in a reported outcome that has a clinical impact.29,32”

Define and distinguish "teaming" from traditional teamwork.

We have defined “teaming” and distinguished it from traditional teamwork in the first paragraph of the Introduction.

“A team can be either bounded, where members have static, defined roles, or dynamic, where members adjust roles continuously based on the task.2,3”

“The dynamic model of teaming is important in healthcare due to the dynamic nature of patient disease trajectories and medical care generally, allowing different members of the healthcare team to adjust their roles as needed to best take care of the patient.4,5”

Explain the significance of family involvement in decision-making.

We have included the following sentence in the Results:

“Caregiver involvement is pertinent because it serves as a way to define quality of care since in the case of serious illness or palliative populations, the patient themselves may have decreased decision making capacity and engagement of the family or caregiver ensures the patient's wishes are respected.”

Ensure accessibility and citation of appendices and supplementary materials.

We have ensured the accessibility and citation of appendices and supplementary materials.

Discussion:

Discuss quality assessment criteria or tools.

We have discussed and defined the quality assessment tool used:

“The Cochrane Risk of Bias tool is an accepted risk of bias assessment for RCTs and considered seven domains: sequence generation, allocation concealment. blinding of participants and personnel, blinding of outcome assessors, incomplete outcome data, selective outcome reporting, and other sources of bias.”

Analyze potential limitations and biases in included studies.

We have included potential limitations and biases of the included studies:

“ The included studies were heterogeneous in nature, which did not allow for statistical analysis, thus limiting the conclusions formed in this study. Finally, the included studies did not provide sufficient information in the manuscript regarding the purpose of surgeon interaction with the patient during the perioperative period (communicate the purpose and expect results of surgery, identify patients who are appropriate for surgery, optimize patients for surgery to minimize risks, or identify and manage complications postoperatively), which may have added an additional layer of understanding.”

The included articles had no overall high risk of bias as per the Cochrane Risk of Bias tool.

Provide specific examples or statistics from reviewed studies.

We have included specific examples from the included studies:

“In the group setting, outcomes in the QLQ-c-30, Profile of Mood States, and MACL were improved, which may be attributed to patients being in the physical presence of each other and providing reassurance and camaraderie, accumulating knowledge together, and asking questions.40,44,48,60,61,62,65,66,77 For example, Koet et al., 2021 demonstrated a statistically significant difference in the QLQ-C30 at the one month follow up between the intervention arm (mean = 72.1) and the control arm (mean = 63.9).44”

“ In almost all the included perioperative studies, longitudinal patient contact was seen, but was especially exemplified by Mertz et al., 2017 where there were no improvements in patient reported outcomes at six months, but there were clinically and statistically significant improvements in the intervention group at twelve months.46”

Include a section on clinical implications of the findings.

We have included clinical implications through the discussion as follows:

“Due to limited resources and time for surgeons, clinical team members can take on an expanded role in cancer care to conduct patient/caregiver education or clinical follow-up with the patients.”

“ Given the well-known workforce constraints for palliative care in cancer, advanced practice providers are an ideal alternative to maintain quality of care while considering overall healthcare expenditure.83 Oncology advanced practice providers can lead follow-up visits and educational sessions for patients in addition to administrative and clinical responsibilities, making this a viable opportunity specifically in the context of perioperative cancer care.”

“Longitudinal follow-up and effective communication among the healthcare team, with referrals to healthcare team members, may be used to improve the quality of life for patients with cancer in the perioperative period.”

“Including caregivers in cancer interventions has also translated to improved patient satisfaction, suggesting it may be a worthwhile component to include in the treatment of patients with cancer in the perioperative period.”

Conclusion:

Specify practical applications and expected benefits of the intervention.

We have included the following statement within the conclusion:

“The intervention components found in this study can be leveraged in efforts to improve perioperative cancer care based on the resources available and the specific needs of the patients.”

Elaborate on the importance of factors like surgeon type and purpose of interaction.

We have included the following statement within the conclusion:

“In addition, different types of surgeons have different practices, workflows, and practice cultures. Interventions that work for one practice culture may need to be adapted to translate into other practice cultures.”

Summarize limitations of the systematic review for transparency.

We have included limitations of the systematic review for transparency:

“Our study can be considered in light of the following limitations. The incorporation of advanced practice providers in the interdisciplinary healthcare team is recent and only three articles that had an intervention with advanced practice providers were included from a limited body of literature. We anticipate an increasing number of publications with interventions that include advanced practice providers. The patient reported outcomes included were heterogeneous and thus a meta-analysis or further synthesis is not possible. Further, PROs hold the bias that patients may have their own individual reference points for answering the questions.95 Our abstraction form did not capture the type of surgeon, which may be relevant in understanding PROs in different perioperative contexts. Furthermore, the search strategy itself, and the screening process, may have missed certain articles. The included studies were heterogeneous in nature, which did not allow for statistical analysis, thus limiting the conclusions formed in this study. Finally, the included studies did not provide sufficient information in the manuscript regarding the purpose of surgeon interaction with the patient during the perioperative period (communicate the purpose and expect results of surgery, identify patients who are appropriate for surgery, optimize patients for surgery to minimize risks, or identify and manage complications postoperatively), which may have added an additional layer of understanding.”

---

## [Editor Report · Decision Letter 2]

5 Nov 2023

Interdisciplinary Interventions that Improve Patient-Reported Outcomes in Perioperative Cancer Care: A Systematic Review of Randomized Control Trials

PONE-D-23-16022R2

Dear Dr. Giannitrapani,

We’re pleased to inform you that your manuscript has been judged scientifically suitable for publication and will be formally accepted for publication once it meets all outstanding technical requirements.

Kind regards,

Sunil Shrestha

Academic Editor

PLOS ONE

Additional Editor Comments (optional):

The authors have addressed all the comments and now it is acceptable.
---

## [Editor Report · Acceptance letter]

10 Nov 2023

PONE-D-23-16022R2 

Interdisciplinary Interventions that Improve Patient-Reported Outcomes in Perioperative Cancer Care: A Systematic Review of Randomized Control Trials 

Dear Dr. Giannitrapani:

I'm pleased to inform you that your manuscript has been deemed suitable for publication in PLOS ONE. Congratulations! Your manuscript is now with our production department. 

Kind regards, 

on behalf of

Dr. Sunil Shrestha 

Academic Editor

PLOS ONE